# Crosslinking-induced patterning of MOFs by direct photo- and electron-beam lithography

Xiaoli Tian[1], Fu Li[1,2], Zhenyuan Tang[2], Song Wang[1], Kangkang Weng[1], Dan Liu[1], Shaoyong Lu[1], Wangyu Liu [1], Zhong Fu[1], Wenjun Li[1], Hengwei Qiu[1], Min Tu [2], Hao Zhang [1] ✉ & Jinghong Li [1,3,4]

Metal-organic frameworks (MOFs) with diverse chemistry, structures, and properties have emerged as appealing materials for miniaturized solid-state devices. The incorporation of MOF films in these devices, such as the integrated microelectronics and nanophotonics, requires robust patterning methods. However, existing MOF patterning methods suffer from some combinations of limited material adaptability, compromised patterning resolution and scalability, and degraded properties. Here we report a universal, crosslinking-induced patterning approach for various MOFs, termed as CLIP-MOF. Via resist-free, direct photo- and electron-beam (e-beam) lithography, the ligand crosslinking chemistry leads to drastically reduced solubility of colloidal MOFs, permitting selective removal of unexposed MOF films with developer solvents. This enables scalable, micro-/nanoscale (≈70 nm resolution), and multimaterial patterning of MOFs on large-area, rigid or flexible substrates. Patterned MOF films preserve their crystallinity, porosity, and other properties tailored for targeted applications, such as diffractive gas sensors and electrochromic pixels. The combined features of CLIP-MOF create more possibilities in the system-level integration of MOFs in various electronic, photonic, and biomedical devices.

MOFs constitute a rich library of materials with myriad geometries, porosities, and functionalities[1–4]. Due to their chemical versatility and high porosity, MOFs are promising in a wide range of applications. Earlier examples mostly exploit MOFs in the bulk, powdery forms for gas storage, selective adsorption, and catalysis[1]. More recently, MOFs in the form of thin films offer disruptive potential to build solid-state devices, with broad implications in next-generation microelectronics, photonics, sensing, and biomedical applications[5–11]. For instance, the ultralow dielectric constants and large ion conduction channels associated with the porous structures render MOFs ideal gap-filling materials in integrated circuits[12] and promising candidates for electrochromic devices[13], respectively. The combination of selective adsorptivity and optical/electrical transduction mechanism via light confinement, diffraction, or luminescence also promises MOFs in photonic chemical and biosensing[14–16].

One of the key challenges to implement MOFs in these thin-film devices, especially those entailing miniaturized components, is the development of material-adapted patterning methods. Ideally, such patterning methods should be (i) scalable to produce high-resolution and uniform patterns; (ii) applicable to MOFs with diverse chemistries, geometries, and functionalities; and (iii) capable to preserve the desirable structural and electrical/optical properties of MOFs for targeted applications. However, patterning methods for MOFs are still in their infancy and cannot meet these criteria[17–19]

[1]Department of Chemistry, Center for Bioanalytical Chemistry, Key Laboratory of Bioorganic Phosphorus Chemistry & Chemical Biology (Ministry of Education), Tsinghua University, Beijing 100084, China. [2]Shanghai Institute of Microsystem and Information Technology, Chinese Academy of Sciences, Shanghai 200050, China. [3]Beijing Institute of Life Science and Technology, Beijing 102206, China. [4]Center for Bioanalytical Chemistry, Hefei National Laboratory of Physical Science at Microscale, University of Science and Technology of China, Hefei 230026, China. ✉e-mail: hzhangchem@mail.tsinghua.edu.cn

(as summarized in Supplementary Table 1). For instance, growing MOFs at selected locations requires substrates with pre-patterned organic monolayers[17–19], lithographically-defined metal electrodes[20–22], or hard/soft templates[23–26]. These requirements lead to complicated patterning procedures and limited adaptability[27]. These methods also suffer from low patterning resolution (tens of µm) and poor film quality. Inkjet[28] and aerosol jet printing[29] of solutions containing metal salts and organic linkers permit scalable in-situ formation and patterning of certain types of MOFs under ambient conditions. However, the printing resolution is generally restricted to over 10 µm due to the complexities in ink rheology and fluidic dynamics. Conventional photolithography is scalable to produce high-resolution patterns due to its parallel nature and affords uniform zeolitic imidazolate framework (e.g., ZIF-8) patterns with ≈10 µm resolution[30]. Unfortunately, it requires photoresists as templates and dry/wet etching steps, which can cause chemical contamination, damage the pore structures, and lead to degraded properties of patterned MOFs[31–33]. To address this, Ameloot and co-workers proposed the direct (or "resist-free") X-ray and e-beam lithography of halogenated ZIF films[31] made from vapor-phase deposition[34]. X-ray or e-beam irradiation triggers a series of halogen radical-mediated structural and chemical changes, which make the exposed ZIFs soluble in developer solvents. The switched solubility after irradiation allows for the selective dissolution of ZIFs in the exposed regions, producing well-defined, micro- and nanoscale patterns. The patterning resolution under e-beam irradiation has reached sub-50 nm while the porous structures are maintained. However, this approach is only applicable to a certain selection of MOFs (namely, halogenated ZIFs) and requires sophisticated apparatus (X-ray and e-beam lithography) that are incompatible with scalable and large-area patterning[32,33,35]. Such limitations in material adaptability and patterning scalability also apply to other e-beam lithography-based MOF patterning methods with comparable patterning resolution[36–40].

Here, we develop a crosslinking-induced patterning approach (termed as CLIP-MOF) that is universal for direct patterning of MOFs with high resolution, quality, and throughput. This method exploits the versatile chemistry of colloidal MOF nanoparticles (NPs)[41–48] and adapts well with the photo- and e-beam lithographic techniques. In CLIP-MOF, smooth MOF films with adjustable thickness are deposited from a colloidal solution containing MOF NPs and a small fraction (as low as 1 wt% to the mass of MOFs) of bisazide-based crosslinkers. Triggered by mild UV exposure (254 nm, 90 mJ cm$^{-2}$) or e-beam irradiation (50 µC cm$^{-2}$), the crosslinkers connect adjacent MOF NPs by covalently bonding their surface ligands. The crosslinked NPs in the exposed area lose their colloidal stability and remain structurally robust during subsequent developing processes, while the unexposed MOFs are dissolved during developing. This enables uniform micro- (≈5 µm, replicating the resolution limit of our photomasks) and nanoscale (≈70 nm) patterning by direct photo- and e-beam lithography, respectively, in the absence of resists. The scalable solution-based film deposition and photolithographic patterning procedures further allow CLIP-MOF for producing large-area MOF patterns (≈10 cm) on rigid and flexible substrates. The patterning is universal to MOFs with diverse chemistries, structures, and properties, benefiting from the nonspecific crosslinking chemistry. Various colloidal MOFs (ZIF-8, ZIF-7, HKUST-1, UiO-66, and Eu(BTC)) can be patterned in a consecutive, layer-by-layer fashion, beyond those achievable in existing methods. Moreover, CLIP-MOF preserves the crystallinity, porosity, and other material properties of patterned MOFs, and permits their uses as diffractive photonic vapor sensors and pixelated electrochromic devices. We expect CLIP-MOF as a versatile and material-adapted approach for integrating MOFs in solid-state devices, with broad implications in microelectronics, nanophotonics, lab-on-chip sensing, and biomedical applications.

## Results

### Patterning mechanism and chemistry

CLIP-MOF via direct, resist-free photo- and e-beam lithography uses the chemistry and procedures shown in Fig. 1. Colloidal MOF NPs (Fig. 1a, b and Supplementary Fig. 1) of various compositions, crystal structures, and sizes (3–60 nm) were synthesized following reported protocols (see "Methods" section) and used as precursors for patterning. These include MOFs with representative structures (ZIF-8, ZIF-7, UiO-66, HKUST-1) and additional functionalities (the luminescent Eu(BTC)). The surface of MOF NPs is coated with a layer of ligands containing abundant C−H groups, whose compositions are listed in Supplementary Table 2. For short, the colloidal MOF NPs are named as (the structure of) MOF@(the composition of) ligands. The ligands support the colloidal stability of MOFs in solvents such as methanol and toluene (Fig. 1b), by providing repulsive forces to outperform the interparticle van der Waals attraction. Such solution-processability of colloidal MOFs was exploited in previous patterning methods via nanoimprinting[49], transfer-printing[50], and evaporation-directed crack-formation[51], which lead to MOF patterns atop templates/stamps and with poorly defined pattern edges. In comparison, CLIP-MOF relies on a different mechanism, namely the ligand crosslinking-induced solubility changes of colloidal MOFs. In brief, light- and e-beam sensitive bisazide-based crosslinkers (e.g., ethylene bis(4-azido-2,3,5,6-tetrafluorobenzoate), or bisPFPA, Fig. 1c and Supplementary Fig. 2) are added to MOF NP solutions without affecting their colloidal stability. The choice of crosslinkers was inspired by our recent work on the direct photo-patterning of colloidal inorganic quantum dots[52–55]. In the coated MOF films, activated by UV exposure (254 nm) or e-beam irradiation, bisPFPA produces nitrene radicals at both ends that undergo nonspecific C−H insertion reactions with ligands on MOF NPs (Fig. 1c). These chemical events can join adjacent MOF NPs, forming crosslinked networks that are insoluble in their original solvents (Supplementary Fig. 3). The drastically reduced solubility of exposed MOF NP films allows for selective dissolution of non-exposed regions during the developing step, and yields patterned films. This mechanism is similar to those of conventional photo- and e-beam lithography, but requires no resists. The patterning procedures are thus compatible with the workflows in lithographic techniques and include only three steps (Fig. 1d, top). (i) Casting films by using solutions containing both colloidal MOF NPs and crosslinkers; (ii) UV or e-beam exposure in selected regions via predefined photomasks or e-beam writing programs; (iii) Developing with suitable solvents to exclusively remove the unexposed MOFs and form micro- and nanoscale patterns with the high-resolution inherent from lithographic techniques. Because the patterns of crosslinked MOFs remain structurally robust against subsequent solution-based processes, these steps can be performed consecutively for multilayered patterning of MOFs with the same or different compositions (Fig. 1d, bottom). More importantly, the nonspecific crosslinking chemistry occurs efficiently on ligands containing C−H groups, regardless of the compositions or structures of MOFs. Thus, CLIP-MOF is universal to a broad scope of colloidal MOFs. This distinguishes it from previous direct X-ray/e-beam patterning methods, which involve the chemical transformation of selective types of MOFs. The combined features render CLIP-MOF scalable, precise, versatile, and non-destructive for direct MOF patterning (Fig. 1e).

We first validated the proposed crosslinking chemistry in CLIP-MOF. Colloidal ZIF-8 NPs capped with polyglycol-based ligands (ZIF-8@BrijC10, Supplementary Fig. 4)[56] were used as a model system. The average size of ZIF-8@BrijC10 NPs is 33 ± 3 nm, as revealed by the dynamic light scattering (DLS) data and transmission electron microscopic (TEM) image (Fig. 2a, b). Colloids containing ZIF-8@BrijC10 and bisPFPA are stable in methanol at a concentration of ≈50 mg mL$^{-1}$. The solution turns turbid after UV exposure due to the formation of large aggregates of crosslinked NPs (sizes >500 nm, Fig. 2a, b). The loss of colloidal stability can be rationalized by the significantly increased van

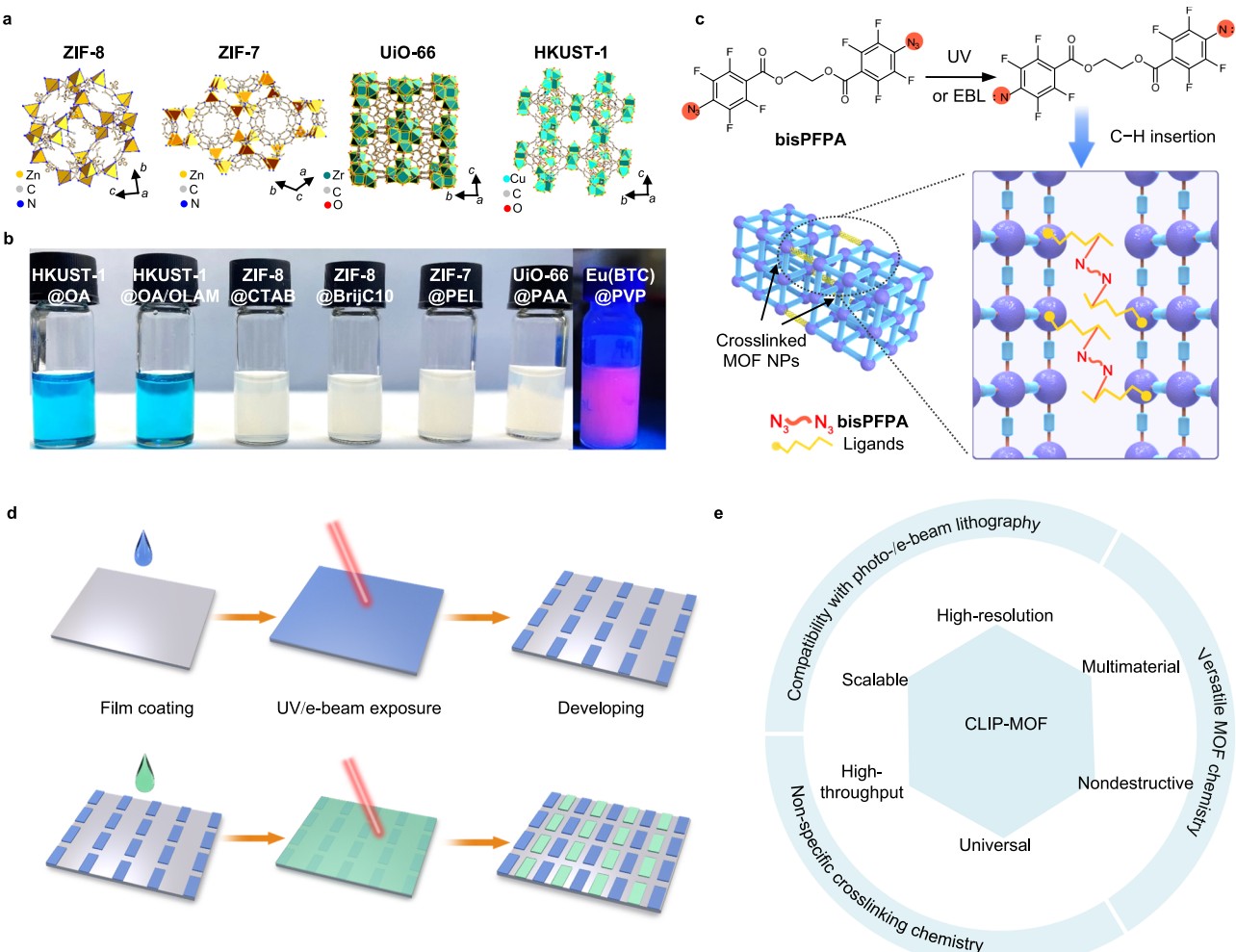

**Fig. 1 | Scheme of the chemistry and procedures of CLIP-MOF. a**, **b** Adaptability to various colloidal MOFs. **a** Crystal structure models of representative MOFs patterned by CLIP-MOF. **b** Photographs of solutions of colloidal MOFs with bisPFPA crosslinkers, as corresponding inks for MOF film coating and patterning. The compositions of the ligands are abbreviated (e.g., OA for oleic acid) here and listed in Supplementary Table 2. The photo of Eu(BTC)@PVP was taken under UV light to show its fluorescence. **c** The crosslinking chemistry for patterning. UV or e-beam irradiation can trigger the decomposition of bisPFPA crosslinkers and generate singlet nitrene radicals at both ends (indicated by red circles). Nitrene radicals then react with the surface ligands via nonspecific C–H insertion. This connects adjacent MOF NPs and yields crosslinked networks that are insoluble in their original solvents. **d** Procedures for CLIP-MOF via direct photo-/e-beam lithography, including film coating, UV/e-beam exposure at desired regions, and selective removal of unexposed MOF films by using developers. The obtained MOF patterns are illustrated by the blue rectangular pixels. This three-step process can be repeated for multilayered patterning of different MOFs, as represented by the blue and green rectangular pixels. **e** Harnessing the versatile colloidal MOF chemistry, efficient and nonspecific crosslinking, and compatibility with well-established lithographic techniques, CLIP-MOF shows a compelling combination of patterning capabilities.

der Waals attractive forces between NP aggregates or the reduced intermolecular entropic changes of crosslinked ligands[57].

The crosslinking events start from the generation of nitrene radicals, followed by their C–H insertion with nearby MOF ligands (e.g., BrijC10). We prepared thin films from a solution of ZIF-8@BrijC10 and bisPFPA (mass ratio, bisPFPA: ZIF-8 ≈ 10 wt%). Fourier-transformed infrared (FTIR) spectra of these films during UV irradiation (254 nm) provide semi-quantitative estimation of the kinetics in nitrene generation. Prior to UV exposure, the films show strong resonance of the azido group (–N₃, asymmetric and symmetric at ≈2100 and ≈1250 cm⁻¹, respectively) of bisPFPA and the C–H moieties from the ligands and linkers of ZIF-8 (2800–3000 and 1300–1500 cm⁻¹) (Fig. 2c and Supplementary Fig. 5). During UV exposure, C–H peak intensities remain almost unaltered and can be used as internal standard. The intensities of –N₃ peaks decline with increased UV doses and diminish at the dose of ≈60 mJ cm⁻². This indicates the fast photolysis of bisPFPA, which converts azides to nitrene radicals. Subsequently, nitrene radicals react with MOF ligands via nonspecific C–H insertion, which form C–N

bonds to connect neighboring MOFs. X-ray photoelectron spectroscopy (XPS) analysis confirms the C–N formation (Fig. 2d). For XPS analysis, we prepared films from solutions containing nitrogen-free, oleic acid capped HKUST-1 NPs (HKUST-1@OA) and bisPFPA. For the CLIP-MOF treated samples, the films were exposed to UV (90 mJ cm⁻²) and developed to remove excess/unreacted bisPFPA crosslinkers. The C1s spectrum of pristine films contains C–C/C=C (284.8 eV) and C–O/C=O (288.6 eV) peaks[36] solely from the linkers and ligands of HKUST-1@OA. The CLIP-MOF treated films show an additional peak for C–N bonds (285.5 eV)[36]. Because we removed unreacted bisPFPA by using developers, this peak should be attributed to the newly formed C–N bonds or C–N bonds from bisPFPA due to crosslinking. The CLIP-MOF treated MOF films also show evident N1s and F1s peaks from relevant moieties in the crosslinkers, which are absent in pristine samples.

### Direct photolithography of MOFs with microscale resolution

The design principle described above endows CLIP-MOF with great patterning capabilities. For example, CLIP-MOF is a resist-free, direct

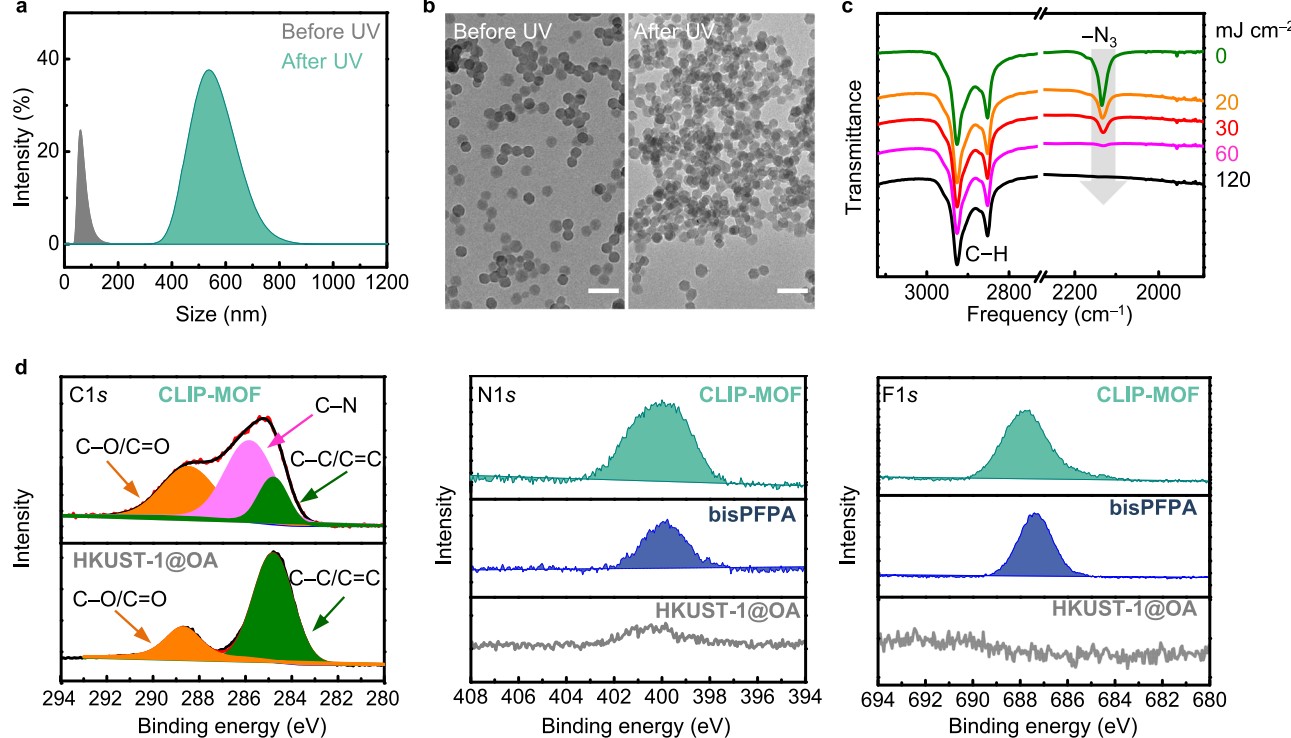

**Fig. 2 | The underlying ligand crosslinking chemistry in CLIP-MOF. a, b** DLS data and TEM images of colloidal ZIF-8@BrijC10 containing bisPFPA crosslinkers (bisPFPA/MOF ≈ 10 wt%) before (gray shaded area) and after (olive shaded area) 254 nm UV irradiation. Scale bars, 100 nm. **c** FTIR Spectra of thin films made from colloidal solution of ZIF-8@BrijC10 with bisPFPA and exposed to UV light with different doses. The asymmetric stretching of –N$_3$ in bisPFPA diminished in the course of 0−60 mJ cm$^{-2}$, indicating the rapid formation of nitrene radicals for subsequent crosslinking. **d** XPS (C1s, N1s, F1s) spectra of film samples of pristine HKUST-1@OA, bisPFPA, and HKUST-1@OA/bisPFPA treated by CLIP-MOF procedures (UV exposure and developing to remove unreacted crosslinkers, labeled as CLIP-MOF). The colored shaded areas in C1s spectra correspond to contribution from different bonds, as indicated by the arrows. Source data are provided as a Source Data file.

photolithographic method for MOFs (Supplementary Table 1). Figure 3 and Supplementary Fig. 6 show the scanning electron microscopic (SEM) and optical images of microscale ZIF-8 patterns in the formats of letters, pixel arrays, and complex pattern, on both rigid and flexible substrates. The patterns were made on spin-coated films from a solution containing ZIF-8@BrijC10 and bisPFPA crosslinkers (the mass ratio of bisPFPA to ZIF-8@BrijC10 ≈ 10 wt%). After UV irradiation (254 nm, 90 mJ cm$^{-2}$), the film was developed by a mixture of methanol/water (vol:vol = 1:1) for ≈5 min. Electron dispersive spectroscopic (EDS) mapping in Fig. 3c confirms that the microdot patterns are composed of crosslinked ZIF-8, where Zn and F are from ZIF-8 and crosslinkers, respectively. The homogeneous elemental distribution suggests uniform MOF patterning. Parameters such as the UV doses strongly affect the patterning quality (Supplementary Fig. 7), correlating well with the trend shown in the FTIR spectra (Fig. 2c). A mild UV exposure of ≈60−90 mJ cm$^{-2}$ is sufficient for the complete conversion of bisPFPA to nitrene, which ultimately leads to patterns with high fidelity. Such UV doses are comparable with those used in commercial organic photoresists. Although the patterns shown in Fig. 3 mostly used ≈10 wt% of crosslinkers, the fraction of crosslinkers can be minimized (as low as 1 wt%) to form high-quality patterns (Supplementary Fig. 8) by using thoroughly washed MOF NPs and extending the developing time to ≈20 min. The finest feature size is ≈5 μm (Fig. 3d), replicating the resolution of the predesigned photomask. Patterning with higher resolution (e.g., ≈1 μm) can be expected by optimizing the lithographic parameters and apparatus. Data from height profiles and line edge roughness (LER) analysis (Supplementary Fig. 9) confirm the uniform and adjustable thickness of patterned ZIF-8 layers. The thickness can be tuned from 200 to 500 nm by controlling the film coating parameters (e.g., spin speed). The LER analysis of these line patterns (10 μm wide) reveals sharp edges with an LER of ≈113 nm, corresponding to the size of three MOF particles. For reference, MOF films patterned via conventional photolithography (e.g., selective etching of MOF films in regions without photoresist protection) show a feature edge of ≈500 nm for ≈10 μm square patterns[30]. Magnified SEM (Fig. 3e) and atomic force microscopic (AFM) images (Fig. 3f) show crack-free, compact films with a low surface roughness of ≈13 nm, similar to those of the as-coated, pristine ZIF-8@BrijC10 films. Furthermore, CLIP-MOF is scalable to make a large-area pattern of the periodic table of elements on a 10 cm silicon wafer (Fig. 3g and Supplementary Fig.10). CLIP-MOF also adapts well to arbitrary substrates (silicon wafer, glass, metal, conductive oxide, etc., Supplementary Fig. 11) in both rigid and flexible formats (Fig. 3h).

We further studied the adhesion stability and film retention curve of patterned MOF films. Note that the substrates used for patterning are not modified physically or chemically. Therefore, the adhesion between patterned MOF films and the substrates come mostly from the van der Waals forces. The van der Waals adhesion force can be estimated by equation[58] and be sufficient to hold MOF films firmly on substrates during the soaking test in different solvents for over 24 h (Supplementary Fig. 12). For more quantitative estimation of the patterning efficacy, we measured the film retention or contrast curve of ZIF-8@BrijC10 MOF films at different UV doses (Supplementary Fig. 13). The film retention reaches ≈83% at the UV dose of ≈90 mJ cm$^{-2}$. From the film retention data, we also estimated the storage/aging time for the solution containing MOF NPs and crosslinkers (Supplementary Fig. 14). Because the patterning relies on the contrast in colloidal solubility of MOFs before and after UV exposure, freshly made MOF solutions with great colloidal stability are preferred and the aging time is about 10 days.

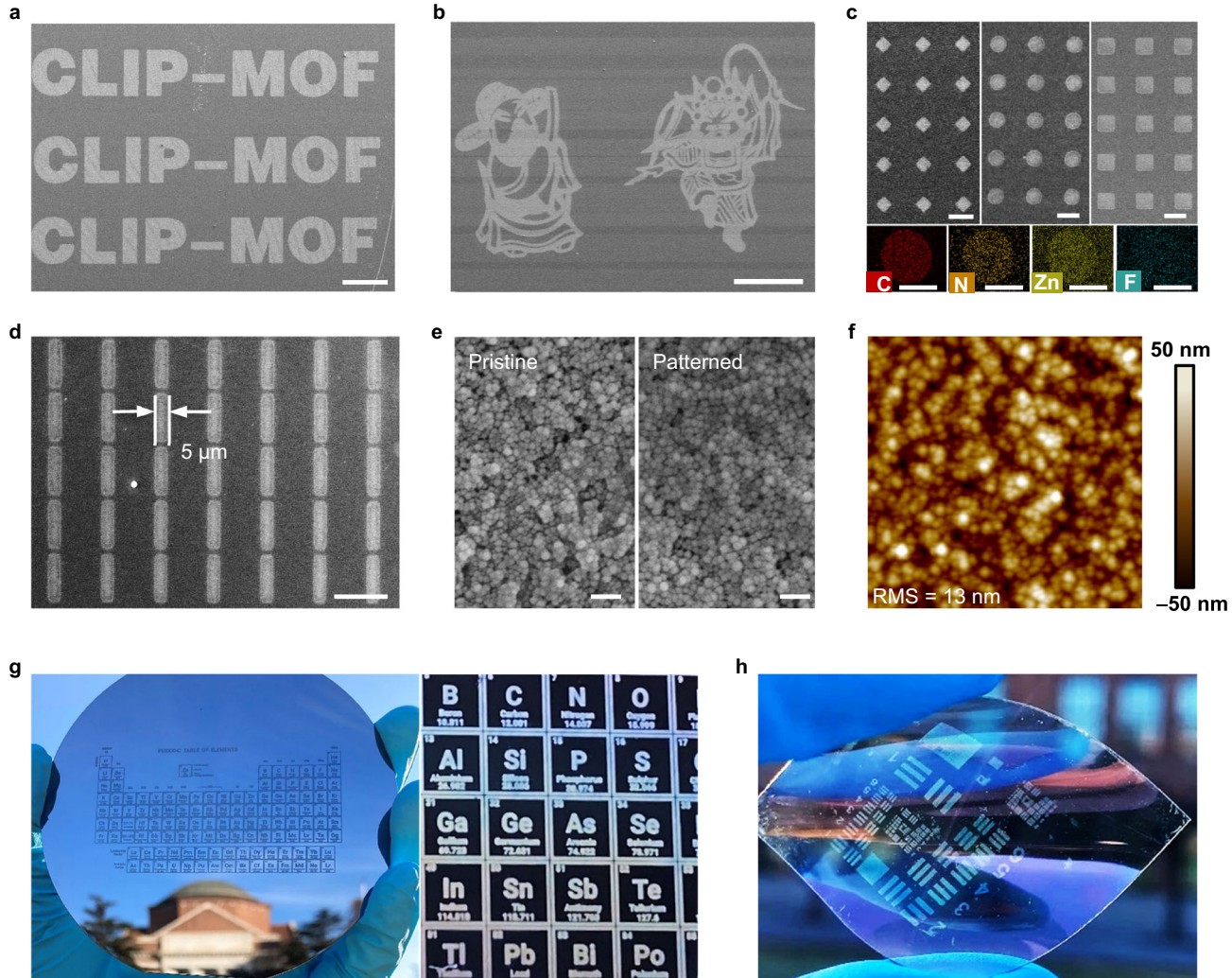

**Fig. 3 | Direct photolithography of ZIF-8@BrijC10. a–d** SEM images of ZIF-8@BrijC10 patterns in the formats of letters, complex pattern, microarrays, and rectangles. The source image for making the pattern in (**b**) is adapted with permission from Visual China (www.vcg.com/creative/1308528815). **c** also shows the EDS colored mapping of patterned microdot arrays for different elements. **e** Top-view SEM images of pristine and patterned MOF films. Scale bars, 200 nm. **f** AFM image of the patterned film. The root mean square (RMS) roughness is 13 nm. **g** Photographs of a large-area pattern of the periodic table of elements on a 10 cm wafer. The zoomed-in view. **h** A photograph of patterned ZIF-8@BrijC10 film on a flexible substrate. The photomask used for **h** has the form of 1951 US Air Force Target, a standard for evaluating patterning quality. Scale bars, **a** 500 μm, **b** 500 μm, **c** 100 μm (SEM) and 50 μm (EDS mapping), **d** 20 μm.

Equally importantly, CLIP-MOF is universal to MOFs with various compositions, structures, porosity, and functionalities. Figure 4a–d shows the patterns of four representative colloidal MOFs, namely the ZIF-8@CTAB, ZIF-7@PEI, HKUST-1@OA/OLAM, and UiO-66@PAA. EDS data in the insets shows the mapping of corresponding metals in each MOF. The patterning parameters for these MOFs appear in Supplementary Table 2. Despite their different material properties, high-fidelity pixel arrays were obtained in all cases. The magnified SEM images of these patterned MOF films (Supplementary Fig. 15) also rule out the possibility of structural damage or amorphization of MOF NPs. MOFs of various compositions, sizes, and surface ligands maintain their well-defined and faceted structures. The expanded scope of patternable MOFs, together with the consecutive patterning capabilities, enables layer-by-layer, multimaterial patterning of different MOFs. Figure 4e and Supplementary Fig. 16 show the optical images and EDS mapping of binary MOF patterns of Zn-containing ZIF-8@BrijC10 and Cu-containing HKUST-1@OA/OLAM as alternating rectangular pixels and cross networks. Due to the robust adhesion of MOF films on the substrates and the significantly reduced solubility of crosslinked MOF NPs, patterning of the second MOF layer does not

affect the underlying crosslinked layer (Supplementary Fig. 16a–c). Despite the small difference in the atomic weight (Zn, ≈65 and Cu, ≈64), the spatial distribution of the two types of MOFs can be recognized in the EDS mapping (Supplementary Fig. 16a). To confirm the stability of the first patterned layer during the subsequent patterning procedures, we further compared the heights of patterned lines in the first layer before and after the patterning of the second layer, as shown in Supplementary Fig. 17. The heights of the first layer remain unchanged, regardless of the compositions of different layers. Furthermore, we made patterns containing three different MOFs (Fig. 4f and Supplementary Fig. 16d, ZIF-8@BrijC10, HKUST-1@OA/OLAM, and ZIF-7@PEI). Prior to this work, there has been only one report showing the selective positioning of two MOFs (UiO-66 and MIL-101) via two orthogonal reaction pathways between surface-functionalized MOF NPs and chemically modified substrates[19]. Patterning of three or more types of MOFs, especially at the resolution of ≈10 μm, has not been shown in previous reports. The above examples highlight the desired combination of scalability, resolution, fidelity, and generality of CLIP-MOF, which are hard to be achieved in existing patterning methods (Supplementary Table 1). This promises the use of

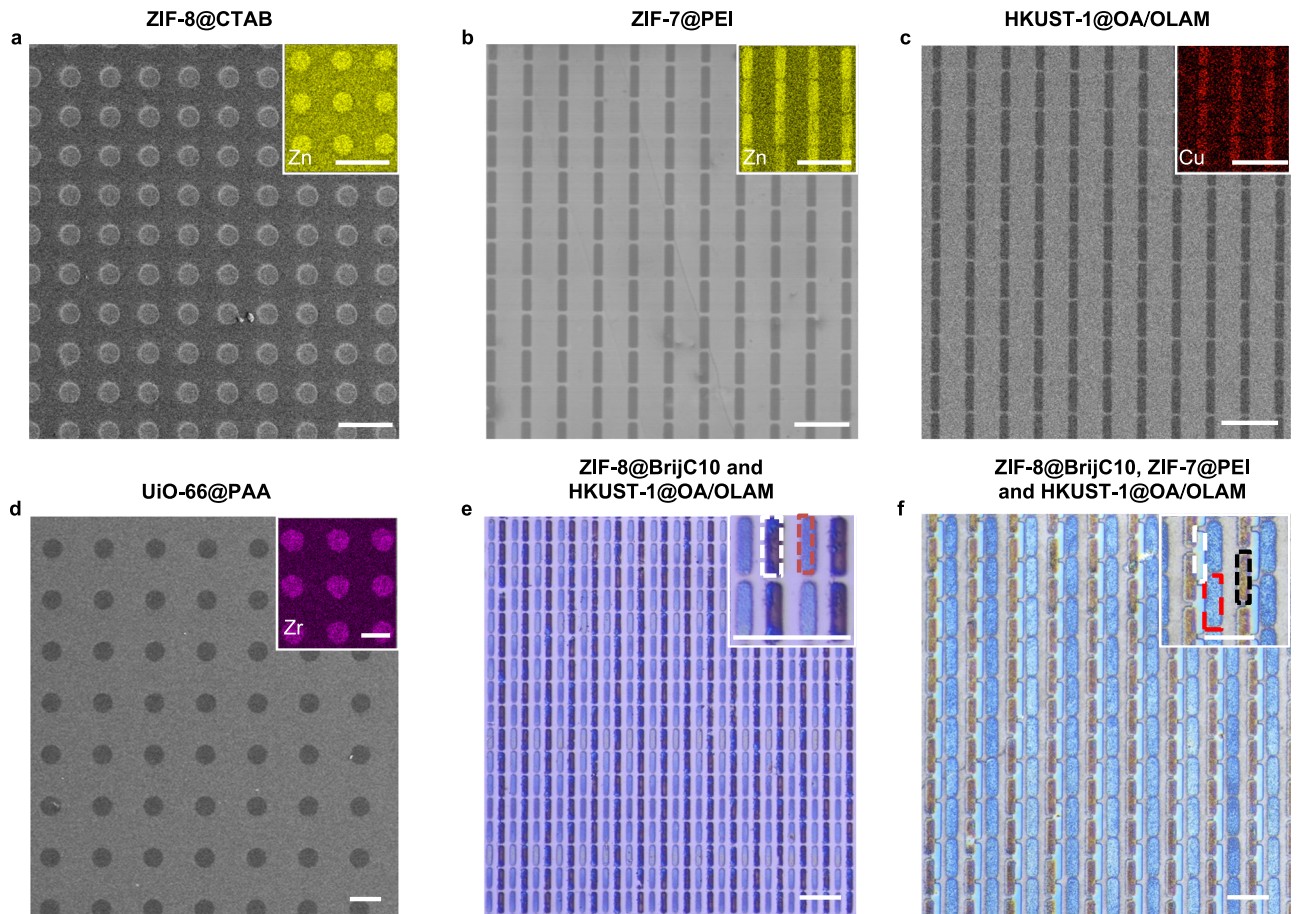

**Fig. 4 | Multimaterial MOF patterning via direct photolithography. a–d** SEM images of patterned ZIF-8@CTAB, ZIF-7@PEI, HKUST-1@OA/OLAM, and UiO-66@PAA. Insets show corresponding EDS data. Scale bars, 100 μm. **e** Optical images of patterns of alternating rectangles composed of Zn-containing ZIF-8@BrijC10 and Cu-containing HKUST-1@OA/OLAM. Inset shows the magnified version highlighting the rectangles in dashed boxes (white, HKUST-1@OA/OLAM; red, ZIF-8@BrijC10). Scale bars, 50 μm. **f** Optical images of patterns containing three different MOFs in the form of rectangular arrays. Inset shows the magnified version highlighting the rectangles in dashed boxes (white, HKUST-1@OA/OLAM; red, ZIF-8@BrijC10; black, ZIF-7@PEI). Scale bars, 50 μm.

CLIP-MOF in incorporating multicomponent MOFs in applications including combinatorial chemical/biosensing and microelectronics/nanophononics.

## Properties of patterned MOFs

CLIP-MOF is nondestructive to the intrinsic properties of MOFs, including the crystallinity, structures, porosity, and coordination bonding, as supported by data from a collection of characterization techniques. As shown in the powder XRD data in Fig. 5a–e and Supplementary Fig. 18, MOF NPs including ZIF-8@BrijC10, ZIF-8@CTAB, ZIF-7@PEI, UiO-66@PAA, HKUST-1@OA and HKUST-1@OA/OLAM, and Eu(BTC)@PVP maintain their crystal structure and crystallinity after being treated by the CLIP-MOF patterning procedures (addition of crosslinkers with the amount of 5–20 wt% to the mass of MOFs, UV exposure, and removal of unreacted crosslinkers by developer solvents). No impurity peaks, additional peak widening, or changes in the relative peak intensities were observed after patterning. Note that the relatively broad peaks of pristine and patterned HKUST-1@OA/OLAM are due to their small particle sizes (about 3 nm, as estimated by Scherrer equation, Supplementary Fig. 18). HKUST-1 synthesized with only OA ligands have larger particle sizes and hence more evident diffraction peaks. Nonetheless, both HKUST-1@OA and HKUST-1@OA/OLAM preserve their structures and crystallinity after patterning. Owing to the high efficiency in the crosslinking chemistry and low UV doses required for patterning, the XRD patterns of patterned MOF films remain identical during UV exposure (up to 1.8 J cm⁻², or

20 times higher than the required dose for patterning), as shown in Supplementary Fig. 19. To validate the crystal structures of patterned MOFs, we further performed Rietveld refinement analysis on XRD data of patterned ZIF-8@BrijC10, ZIF-7@PEI and UiO-66@PAA films, where the measured data matches well with the calculated diffraction patterns (Fig. 5f–h). In brief, patterned ZIF-8@BrijC10 exhibits a cubic phase with the I−43 m space group and lattice parameters $a = b = c = 17.0886(55)$ Å, $\alpha = \beta = \gamma = 90°$. These parameters are consistent with those reported for ZIF-8[59]. Similarly, patterned ZIF-7@PEI and UiO-66@PAA samples exhibit a hexagonal phase (R−3 space group)[59] and a cubic phase (Fm−3m space group)[60], respectively, consistent with reported phases. Complementary to the XRD data analysis, the selected area electron diffraction (SAED) patterns from high-resolution TEM (HRTEM) analysis provide microscopic information of the crystallinity of MOFs. Figure 5i, j shows evident and sharp diffraction spots for both pristine and patterned ZIF-8@CTAB samples, indicating the preservation of their crystallinity at microscale. We also captured the HRTEM and corresponding EDS data of these samples (Supplementary Fig. 20). MOF NPs in both cases show well-defined shapes and homogeneous distribution of Zn (from the metal ions) and N elements (from the organic linkers). Fluorine only exists in the patterned sample due to the crosslinking reaction.

The unchanged framework results in almost identical nitrogen adsorption and desorption isotherms of pristine and treated samples (Fig. 5k–n and Supplementary Table 3). For instance, the estimated Brunauer-Emmett-Teller (BET) surface area of pristine and patterned

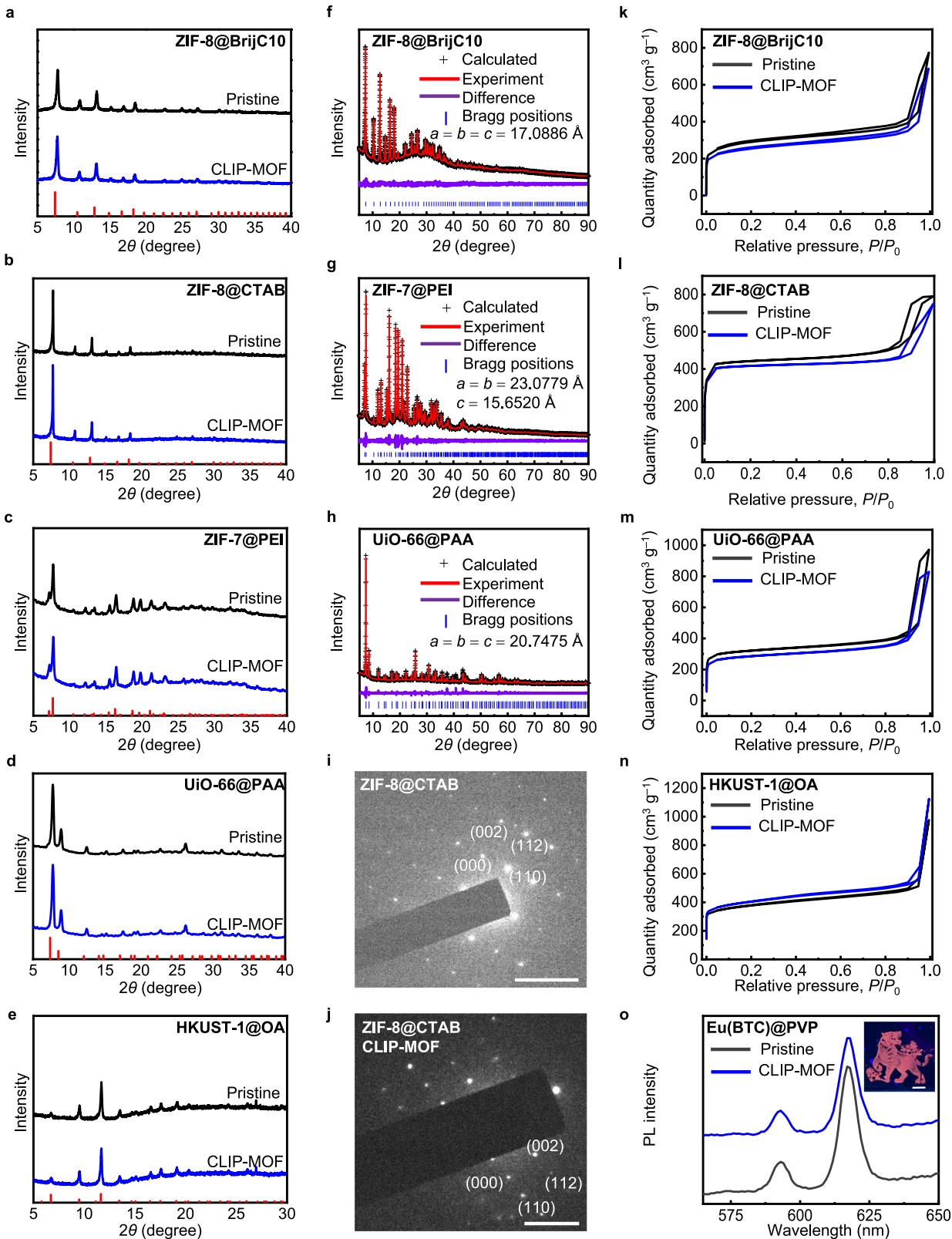

ZIF-8@CTAB NP MOF are 1721.3 and 1660.3 cm$^2$ g$^{-1}$, respectively. These values are on par with those reported for colloidal ZIF-8 MOFs (900–1630 cm$^2$ g$^{-1}$)[61]. The pristine and patterned MOFs also show similar pore size distribution (Supplementary Fig. 21). The fully preserved porous properties follow from the absence of resist contamination or plasma/etchant damages, which are the major concerns for conventional photolithography of MOFs[33].

In addition to the preserved crystallinity and porosity, we also confirmed that the bonding between metal ions and organic linkers in MOFs remain intact before and after patterning. Supplementary Fig. 22 compares the FTIR spectra of pristine MOFs films, MOF films containing bisazide crosslinkers before UV exposure, and MOF films treated by CLIP-MOF procedures. The pristine samples show resonance related to C−H bonds (at 2800–3000 cm$^{-1}$) in the organic linkers

**Fig. 5 | Preserved crystallinity and porosity of patterned MOF films. a–e** Powder XRD data of CLIP-MOF patterned films (blue traces) of **a** ZIF-8@BrijC10, **b** ZIF-8@CTAB, **c** ZIF-7@PEI, **d** UiO-66@PAA, **e** HKUST-1@OA. For comparison, XRD patterns of pristine MOF films (black traces) and the standard diffraction patterns (indicated by red vertical lines at the bottom) are included. **f–h** Rietveld refinement of powder XRD data for CLIP-MOF treated **f** ZIF-8@BrijC10, **g** ZIF-7@PEI, and **h** UiO-66@PAA powders. The measured and calculated data are shown in red lines and the plus marks, respectively, with their differences plotted in purple lines. The blue vertical bars indicate the allowed peak positions. Fitted lattice parameters are also included. Goodness of data fitting, **f** Rp = 2.8% and Rwp = 3.26%, **g** Rp = 4.87% and Rwp = 3.79%, **h** Rp = 2.02% and Rwp = 5.58%. Rp stands for the full spectrum factor

and Rwp stands for the weighted full spectrum factor. **i, j** SAED patterns of pristine and CLIP-MOF treated ZIF-8@CTAB films. SAED patterns along the [1–10] zone axis. Scale bars, 2 nm$^{-1}$. **k–n** $N_2$ adsorption–desorption isotherms of various pristine (black traces) and CLIP-MOF treated MOF NP powders (blue traces). **k** ZIF-8@BrijC10, **l** ZIF-8@CTAB, **m** UiO-66@PAA, **n** HKUST-1@OA. **o** PL spectra of pristine (black traces) and patterned (blue traces) Eu(BTC)@PVP films. Inset is the photograph of patterned Eu(BTC)@PVP, taken under UV light. The source image for making the pattern in (**o**) is adapted with permission from Visual China (www.vcg.com/creative/1352846001). Scale bar, 2 mm. Source data are provided as a Source Data file.

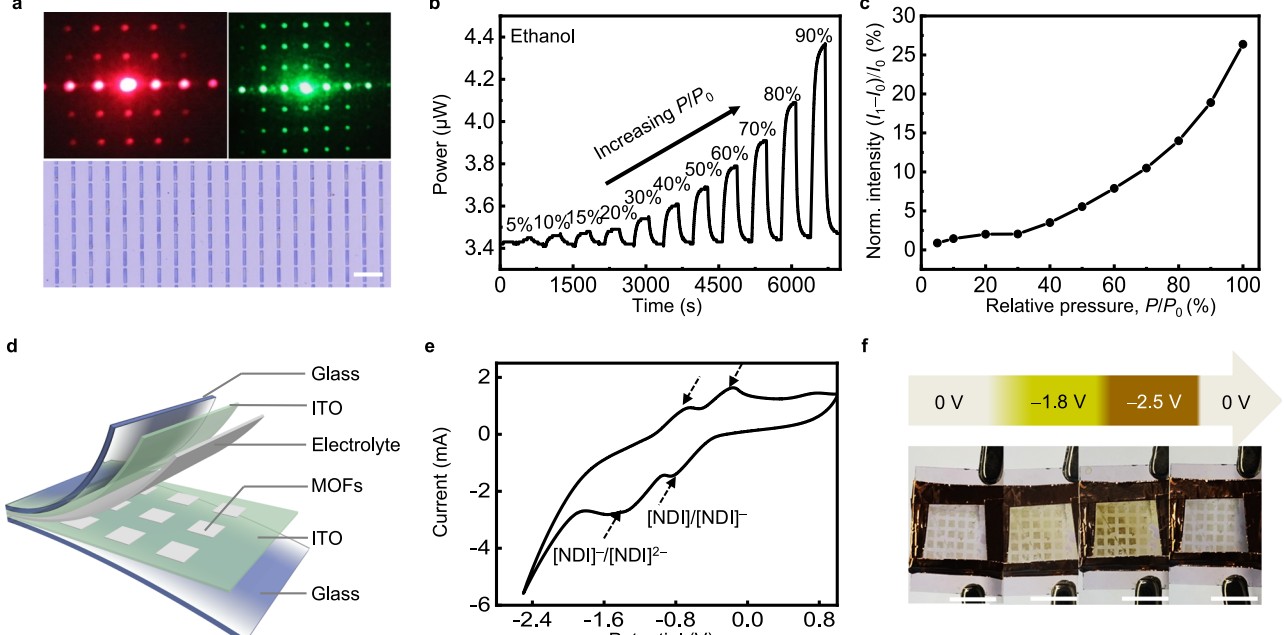

**Fig. 6 | Example applications of patterned MOF films. a** Photographs of diffraction patterns obtained by illuminating 2D grating structures of ZIF-8@BrijC10 on transparent quartz substrate with red and green lasers. **b** Responses of the patterned ZIF-8@BrijC10 diffraction grating towards ethanol vapors. The measured optical power corresponds to the light intensity of the first-order diffraction spot, which changes with the relative vapor pressure ($P/P_0$) of ethanol (indicated as percentage numbers). **c** The normalized first-order diffraction spot intensity (($I_1$–$I_0$)/$I_0$) of the grating sensors as a function of $P/P_0$ of ethanol. $I_0$ is the intensity

measured at $P/P_0 = 0\%$. **d** Scheme of the solid-state, electrochromic devices with a patterned MOF layer sandwiched between two conducting electrodes filled with electrolytes. The MOF layer of ZIF-8@BrijC10 contains adsorbed electrochromic, naphthalene diimide (NDI) molecules. **e** Cyclic voltammetry curve of the NDI molecules. The dashed arrows indicate the redox peaks. **f** Photographs of the pixelated electrochromic device operated at different potential. Scale bars, **a** 50 μm, **f** 1 cm. Source data are provided as a Source Data file.

as well as bonds between metal ions and the organic linkers. For instance, pristine ZIF-8 or ZIF-7 MOFs show strong Zn–N band at 421 cm$^{-1}$ [62]. The addition of bisazide-based crosslinkers to pristine MOFs introduces additional peaks corresponding to the azido group (e.g., ≈2100 cm$^{-1}$). For ZIF MOF samples treated with CLIP-MOF procedures, resonance from the azido group disappeared while that for Zn–N band remain unchanged. Meanwhile, no additional peaks are observed at 1843 and 791 cm$^{-1}$ for ZIFs, which excludes the potential degradation process involving the N–H groups[62]. Similarly, the preservation of Cu–O band at 492 cm$^{-1}$ and Zr–O band at 663 cm$^{-1}$ for HKUST-1[63] and UiO-66[46], respectively, confirms the retention of chemical bonding and structures of these patterned MOFs. Patterned MOFs also retain their optical properties. The lanthanide-containing MOF (Eu(BTC)@PVP) films remain luminescent after patterning, and show the same emission features with the pristine samples (Fig. 5o).

### Examples of the applications of patterned MOF films

Patterning can also introduce structure-defined functionalities beyond those achievable in the constituents. One of the representative

examples is the photonic systems that incorporate the original properties of MOFs (porosity and adsorption selectivity) and the emergent features from the periodic structures (light confinement, diffraction, and guiding)[49]. Figure 6a shows the diffraction patterns of ZIF-8 films with a 2D grating geometry, when illuminated with red and green laser pointers. As a proof of concept, the diffracting platform (total area, 2.5 × 2.5 mm$^2$; linewidth, 5 μm; pitch size, 25 μm) was tested as a photonic vapor sensor. The signal transduction is based on the changes in the refractive indices upon the adsorption of volatile molecules (e.g., ethanol and acetone for ZIF-8), which results in changes in the first-order diffraction intensity[31,49]. Such diffractive grating sensors can monitor the vapor pressure increases of ethanol and acetone in a closed chamber (Fig. 6b, c and Supplementary Fig. 23). We also tested the stability of such sensors in multi-cycle treatment and gas sensing (see "Methods" section). During this multi-cycle utilization, the optical power of first-order diffraction spot remained essentially unchanged (Supplementary Fig. 24a). After these tests, we measured the SEM images of the grating (Supplementary Fig. 24b, c), where the MOF patterns were intact with well-defined facets.

Patterned MOFs with electrochromic linker molecules or adsorbents were also made with potential use in smart displays[13,64]. We mixed N,N-bis(5-isophthalic acid) naphthalene diimide (NDI) with ZIF-8@BrijC10 colloids, during which part of the NDI molecules were bound to the MOF NP surface and the unbound ones were removed by rinsing. The resultant MOF NPs were then added with bisPFPA crosslinkers and patterned via CLIP-MOF. To build a solid-state pixelated electrochromic device (Fig. 6d), the patterned MOF layer was sandwiched between two transparent conductive electrodes (ITO-coated glass), followed by the addition of a small volume of electrolyte and device encapsulation. Cyclic voltammetry test of this device shows evident redox peaks of the NDI molecules (Fig. 6e and Supplementary Fig. 25), indicating the successful incorporation of NDI in the patterned MOF films. The two quasi-reversible processes at $E_{1/2} = -0.85$ and $-1.55$ V are consistent with the reported [NDI]/[NDI]$^-$ and [NDI]$^-$/[NDI]$^{2-}$ redox couples, respectively[13,65]. When operated at low potential, the color of pixels (1 mm × 1 mm) changes from almost transparent (zero bias) to yellow (−1.8 V) and dark yellow or brown (−2.5 V), as shown in Fig. 6f. These color changes are fast (typically within 2 s) and reversible.

### Direct e-beam lithography of MOFs with nanoscale resolution

Beside the microscale patterning via direct photolithography, CLIP-MOF also supports nanoscale patterning by using e-beam lithographic apparatus. Nanoscale patterning is critical for the integration of MOFs in nanophotonics and miniaturized electronics, such as the use of patterned MOFs as the low $k$ gap-filling materials between nanoscale metal interconnects in integrated circuits[12]. Encouraging progress in resist-free, direct e-beam lithographic patterning of MOF films has been made by several groups during the past three years[31,36–40,66]. These approaches rely on e-beam irradiation-induced chemical and structural changes in MOFs, including amorphization[37,66] and disintegration by halogen radicals[36]. These chemical/structural modifications lead to solubility changes of irradiated MOFs in certain solvents and support MOF patterning. The patterning resolution can achieve sub-50 nm. However, these methods usually require high areal doses (1–20 mC cm$^{-2}$ and sometimes additional thermal treatments) and/or specific MOF compositions (e.g., halogen-containing organic linkers). These limit the scope of patternable MOFs to mostly ZIFs and lead to low patterning efficiency due to the long exposure time.

In CLIP-MOF, the changes in MOF solubility originate from the efficient ligand crosslinking chemistry with bisPFPA. It does not require amorphization or disintegration of MOFs and thus permits patterning at 2–3 orders of magnitude lower e-beam doses (e.g., 50 μC cm$^{-2}$). Figure 7a shows the SEM images of patterned stripes of HKUST-1@OA/OLAM. The smallest linewidth is ≈70 nm, on par with the highest resolution achieved in existing direct e-beam lithography of MOFs[30]. The patterns show uniform heights (≈70 nm) defined by the thickness of spin-coated MOF films, as revealed by the AFM topologies and height profiles in Fig. 7b, c, and Supplementary Fig. 26. The LER analysis of these line patterns (130 nm wide, 70 nm thick) reveals sharp edges with an LER of ≈1.5 nm. The surface roughness of patterned films is as low as ≈2.3 nm (Fig. 7d), which corresponds to the diameter of a single HKUST-1@OA/OLAM NP (≈3 nm). Note the ligands used to for colloidal MOFs, e.g., OA, can also self-crosslink under e-beam irradiation without additional crosslinkers[67]. However, these self-crosslinking reactions involve the cleavage of C−H bonds (bond energy ≈300 kJ mol$^{-1}$) and the formation of new C=C bonds, which require much larger energy than the decomposition of azide groups in bisPFPA crosslinkers. With bisPFPA crosslinkers, the patterning requires only ≈50 μC cm$^{-2}$. Without crosslinkers, we obtained no patterns even at 200 μC cm$^{-2}$ (Fig. 7e). This contrast corroborates the proposed patterning chemistry. In addition to its high efficiencies, CLIP-MOF uses films coated from corresponding colloidal MOF NPs, which can be extended to pattern different MOFs. The current lateral resolution (≈70 nm) can also be further improved by

optimizing the patterning parameters, crosslinking chemistry, and the colloidal properties of MOFs.

## Discussion

In conclusion, the presented CLIP-MOF method enables resist-free, direct patterning of MOFs with various compositions, structures, and properties. The patterning mechanism, namely the ligand crosslinking-induced solubility changes of colloidal MOFs, is tailored for the versatile MOF chemistry and compatible with the photo- and e-beam lithographic techniques. Patterned MOF films show preserved structures and properties desirable for applications in photonic sensors and luminescent/electrochromic pixels for smart displays. These features lead to a compelling combination of patterning scalability, resolution, quality, and material adaptability in CLIP-MOF, which are beyond those achievable in existing patterning methods.

CLIP-MOF benefits from the interplay between the colloidal stability and the ligand chemistry of colloidal MOFs. Further optimizations can broaden the scope of patternable MOFs and increase the efficacy of CLIP-MOF. On the one hand, the expanding library of colloidal MOFs and ligands[43] suggests that a vast number of MOFs can be patterned via CLIP-MOF due to the nonspecific crosslinking chemistry, which can provide diverse functionalities in related applications. One should note that the use of colloidal MOF NPs is also associated with challenges and opportunities related to the small crystal size, defects, and the packing fashion. Recent advancements in colloidal MOF NPs[43,60,68–71] allow CLIP-MOF to benefit from nanoscale MOFs with not only decent porosity and surface area but also distinctive features for biosensing and catalysis, such as defect-related catalytic activity and packing-induced mesoporous structures to host macromolecules. The sizes of MOF NPs also affect the surface roughness of patterned MOFs in CLIP-MOF; smaller NPs with uncompromised porosity are preferred for patterning smooth films. On the other hand, crosslinkers and relevant ligand chemistry have been proven crucial in the direct patterning of other colloidal inorganic nanomaterials[53,54,72]. The rational design of crosslinkers and other photo-/e-beam sensitive chemicals that interact with colloidal MOFs in different fashions can support more patterning mechanism and capabilities for MOFs and boost their integration in various device platforms.

## Methods
### Chemicals

The synthesis of colloidal MOF NPs used the following chemicals. Zinc nitrate hexahydrate (Zn(NO$_3$)$_2$·6H$_2$O, 99.99%, Aladdin), copper nitrate trihydrate (Cu(NO$_3$)$_2$·3H$_2$O, 99.5%, Adamas), europium nitrate hexahydrate (Eu(NO$_3$)$_3$·6H$_2$O, AR, Macklin), zirconyl chloride octahydrate (ZrCl$_2$·8H$_2$O, 99.9%, Aladdin), terephthalic acid (99%, Adamas), 2-methylimidazole (97%, Alfa Aesar), benzene-1,3,5-tricarboxylic acid (H$_3$BTC, 98%, Meryer), oleic acid (OA, tech. 90%, Alfa Aesar), oleylamine (OLAM, C18: 80–90%, Acros), benzimidazole (98%, Meryer), hexadecyl trimethyl ammonium bromide (CTAB, 98%, TCI), polyethylene glycol hexadecyl ether (BrijC10, average $M_n$≈683 g mol$^{-1}$, Aladdin), polyethyleneimine (PEI, Sigma Aldrich, average $M_w$≈ 25,000 g mol$^{-1}$), polyvinyl pyrrolidone (PVP, $M_w$≈58,000 g mol$^{-1}$, Harveybio), polyacrylic acid (PAA, $M_w$≈2000 g mol$^{-1}$, Aladdin), sodium acetate trihydrate (CH$_3$COONa, 99%, Alfa Aesar), acetic acid (CH$_3$COOH, Greagent, 99.5%), NaOH (99%, Meryer), dimethyl sulfoxide (DMSO, ≥99.5%, Alfa Aesar), N,N-dimethyl formamide (DMF, ≥99.7%, Alfa Aesar), methanol (99.8%, Acros), ethanol (99.5%, Fisher), hexane (99.5%, Fisher), cyclohexane (Macklin, AR), toluene (99.5%, Fisher). OA and OLAM were degassed prior to use in the glovebox. All other chemicals were used as received.

For the synthesis of bisPFPA crosslinkers: methyl 2,3,4,5,6-pentafluorobenzoate (97%, Heowns), sodium azide (NaN$_3$, J&K Chemicals), 4-(dimethylamino) pyridine (DMAP, 98%, Energy Chemical), N′-ethyl-N-(3-(dimethylamino)propyl) carbodiimide hydrochloride (EDC, 99%,

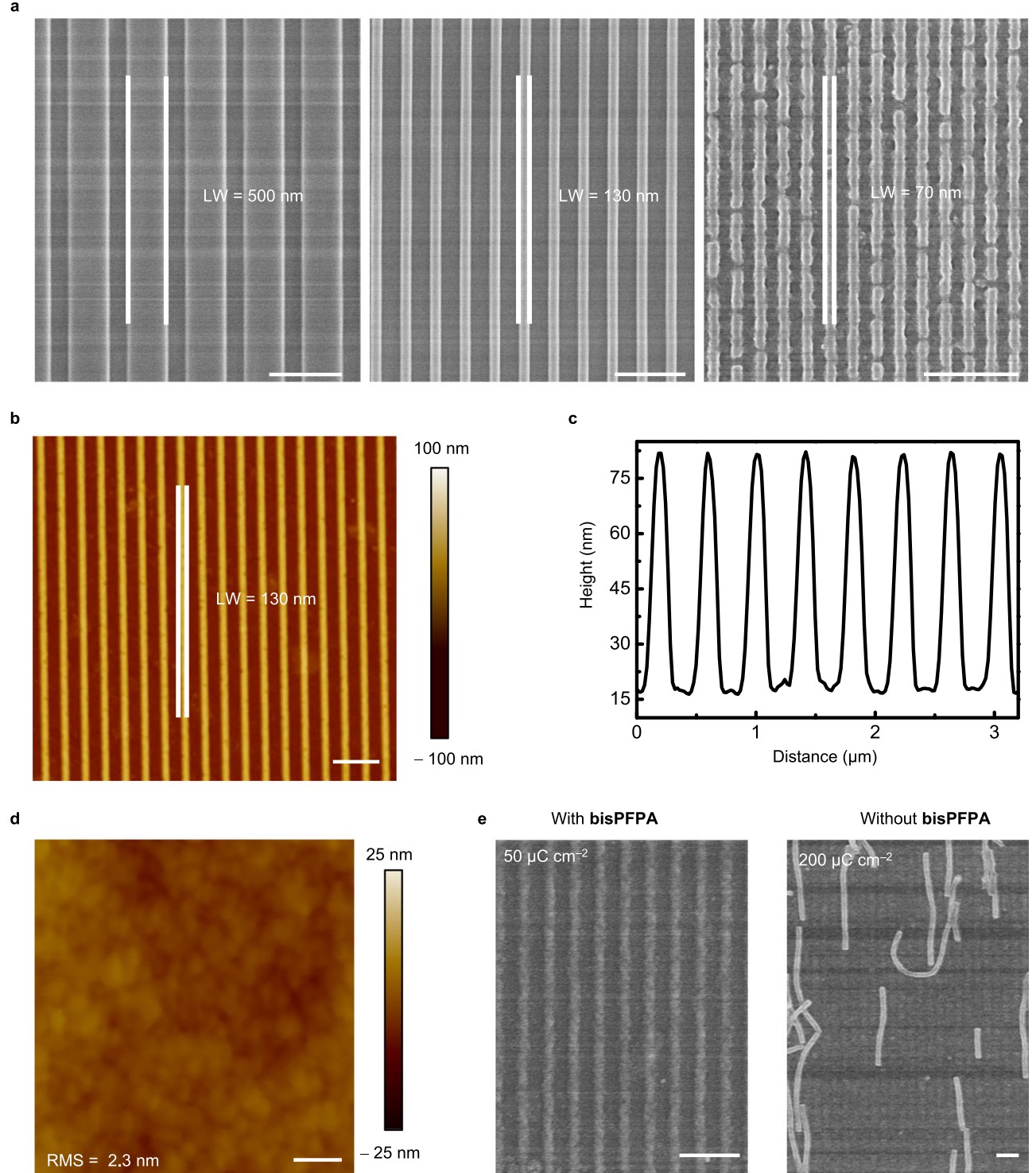

**Fig. 7 | Direct e-beam lithography of HKUST-1@OA/OLAM. a** SEM images of patterned stripes, with the finest linewidth (LW) of ≈70 nm (e-beam dose, 80 μC cm⁻²). **b**, **c** AFM image and height profiles of patterns with 130 nm linewidth. **d** AFM topology showing a minimal surface roughness of ≈2.3 nm. Scale bar, 50 nm.

**e** SEM images of HKUST-1@OA/OLAM patterns with 130 nm linewidth. Well-defined lines were patterned at a low e-beam dose (50 μC cm⁻²) with crosslinkers. No patterns were formed in the absence of crosslinkers, even at higher doses (200 μC cm⁻²). Scale bars, 1 μm. Source data are provided as a Source Data file.

J&K Chemicals), magnesium sulfate (98%, Innochem), sodium hydroxide (99%, Innochem), hydrochloric acid (37% solution in water, Tongguang Fine Chemicals), ethylene glycol (99%, Innochem), chloroform (90%, Innochem), dichloromethane (AR, Innochem), acetone (AR grade, General Reagent), diethyl ether (AR grade, General Reagent), methanol (AR grade, General Reagent), deionized water

(General Reagent, 18.2 MΩ cm resistivity), silica gel (100–200 mesh, Innochem). All chemicals were used as received.

## Synthesis of colloidal MOF NPs

All types of colloidal MOF NPs were synthesized according to reported protocols with some modifications. Note the colloidal stability of these

NPs gradually decreased during storage. Freshly synthesized NPs were preferred to obtain high-quality patterns.

### ZIF-8@BrijC10[56]

BrijC10 (3.420 g) and an aqueous solution of 2-methylimizadole (1.0 mL, 4.0 M) were added to cyclohexane (15.0 mL) under stirring at 37 °C. The mixture formed a reverse micelle solution. In another flask, a mixture of BrijC10 (3.420 g) and an aqueous solution of $Zn(NO_3)_2 \cdot 6H_2O$ (1.0 mL, 1.0 M) was added to cyclohexane (15.0 mL) under the same condition. The resultant solution was added to the solution containing BrijC10 and 2-methylimizadole under vigorous stirring. The reaction last about 2 h. For purification, ethanol (30 mL) was added to the crude solution. ZIF-8@BrijC10 NPs were separated by centrifugation at $12,800 \times g$ for 3 min. The collected NPs were then washed with ethanol once and redispersed in methanol (50 mg mL$^{-1}$) for further use.

### ZIF-8@CTAB

ZIF-8@CTAB NPs were synthesized following ref. 73 with slight modifications. CTAB (0.792 g) was dissolved in a solution of $Zn(NO_3)_2 \cdot 6H_2O$ (2.933 g) in methanol (200 mL). The mixture was then added to a solution of 2-methylimidazole (6.489 g) in methanol (200 mL) under stirring at room temperature. After 1 h, ZIF-8@CTAB NPs were separated as precipitates by centrifugation ($12,800 \times g$, 10 min) and washed with methanol twice. The purified NPs were dispersed in methanol at a concentration of 50 mg mL$^{-1}$.

### HKUST-1@OA[63]

Aqueous solution of NaOH (0.8 mL, 0.125 M), ethanol (1.2 mL), OA (0.3 mL), and hexane (0.17 mL) were mixed under stirring at 50 °C to form a microemulsion. Then an aqueous solution of $Cu(NO_3)_2 \cdot 3H_2O$ (0.2 mL, 50 mg mL$^{-1}$) was added to the microemulsion. The solution was stirred for 10 min, followed by the addition of a mixture of a $H_3BTC$ solution in ethanol (0.17 mL, 60 mg mL$^{-1}$) and deionized water (0.13 mL). The mixture was stirred at 70 °C for 2 h. The resultant HKUST-1@OA NPs were separated from the crude solution by centrifugation at $6800 \times g$ for 3 min. NPs were washed with a mixture of cyclohexane and ethanol (1:1 in volume) for three times, and redispersed in methanol (50 mg mL$^{-1}$) for further use.

### HKUST-1@OA/OLAM[74]

The synthesis started from the preparation of Cu(II)-oleate precursors. Sodium oleate (20 mmol) was dissolved in a mixture of $H_2O$ (40 mL) and ethanol (20 mL) by vortex. An aqueous solution of $Cu(NO_3)_2$ (1.0 M, 10 mL) was quickly added to the above solution, producing green precipitates during stirring. The waxy solids were separated from the supernatant and washed by a mixture of water and ethanol (1:1 in volume). The obtained Cu (II)-oleate was redispersed in cyclohexane (100 mL), forming a transparent green solution (0.1 M). Residual water and insoluble parts were removed by centrifugation ($1700 \times g$, 3 min).

The synthesis of HKUST-1@OA/OLAM NPs used two solutions. Solution A: OLAM (0.7 mmol) was added to the Cu (II)-oleate solution (8 mL), forming a deep blue solution. Solution B: $H_3BTC$ (1.88 mmol) was dissolved in a mixture of DMSO (0.2 mL) and ethanol (0.8 mL). Solution B was then injected in solution A under vigorous stirring at room temperature for 30 min. The produced NPs were precipitated by adding ethanol (8 mL) and collected by centrifugation ($12,800 \times g$, 1 min). The washing process was repeated twice. The purified NPs were redispersed in toluene (55 mg mL$^{-1}$).

### ZIF-7@PEI[75]

$Zn(NO_3)_2 \cdot 6H_2O$ (0.446 g) and benzimidazole (0.354 g) were added into a DMF solution of branched PEI (200 mL, 0.7 mg mL$^{-1}$) under stirring. After being kept at room temperature for 24 h, the same volume of toluene was added to the solution. The produced ZIF-7@PEI NPs were then collected as solids after centrifugation. NPs were redispersed in methanol at a concentration of 55 mg mL$^{-1}$.

### UiO-66@PAA[76,77]

$ZrCl_2 \cdot 8H_2O$ (210 mg) was dissolved in DMF (30 mL). A solution of terephthalic acid (500 mg) in DMF (10 mL) was added to the above solution. Then an aqueous solution of acetic acid (2.6 M, 96 μL) was added to the mixture. The solution was heated at 90 °C for 18 h under stirring. The bare UiO-66 NPs were collected after centrifugation ($12,800 \times g$, 5 min) and suspended in water. UiO-66 NPs (50 mg mL$^{-1}$) were redispersed in a methanol solution of PAA (1 mL, 100 mg mL$^{-1}$) and stirred for 2 h. This reaction yielded PAA@UiO-66 NPs, which were collected after centrifugation and redispersed in methanol (50 mg mL$^{-1}$).

### Eu(BTC)@PVP[78]

$H_3BTC$ (0.0028 g) and $Eu(NO_3)_3 \cdot 6H_2O$ (0.0125 g) were dissolved in 30 mL of DMF as the reaction precursors. PVP (0.0365 g) and $CH_3COONa \cdot 3H_2O$ (0.0229 g) were dissolved in a mixture of ethanol (4 mL) and $H_2O$ (4 mL). The precursor solution in DMF was added to the latter solution. The reaction took 24 h at 80 °C under stirring. Eu(BTC)@PVP NPs were separated by centrifugation ($6800 \times g$, 5 min) and redispersed in methanol (55 mg mL$^{-1}$) for further use.

### Synthesis of bisPFPA crosslinkers

bisPFPA crosslinkers were synthesized according to reported methods[79,80], as shown in Supplementary Fig. 2. Under stirring, $NaN_3$ (1.50 g) and methyl 2,3,4,5,6-pentafluorobenzoate (4.86 g) were added to a mixture of acetone (40 mL) and water (15 mL). The mixture was refluxed for 8 h and placed under low vacuum to remove organic solvents at room temperature. The obtained liquid was added with water (30 mL) and extracted with diethyl ether (3 × 50 mL). The extract was dried over anhydrous $MgSO_4$, followed by rotary evaporation to remove organic solvents, yielding a red liquid of methyl 4-azido-2,3,5,6-tetrafluorobenzoate. For the next step of synthesis, NaOH (7 mL, 20% aqueous solution) was added to methyl 4-azido-2,3,5,6-tetrafluorobenzoate (4.99 g) in methanol (85 mL). The solution was stirred for 12 h at 25 °C. The solution was then acidified by 2 N HCl in an ice bath until pH <1 and extracted by $CHCl_3$ (3 × 50 mL). The extract was dried over anhydrous $MgSO_4$. Methyl 4-azido-2,3,5,6-tetrafluorobenzoic acid was obtained as a colorless solid after solvent evaporation. For the last step of the synthesis, a solution of DMAP (0.13 g) and methyl 4-azido-2,3,5,6-tetrafluorobenzoic acid (2.50 g) in dry $CH_2Cl_2$ (50 mL) was stirred with ethylene glycol (0.33 g) under nitrogen at room temperature for 30 min. After the addition of EDC (2.27 g), the solution was stirred overnight at room temperature. The solution was then added with water (30 mL) and stirred for another 30 min. The mixture was extracted with $CH_2Cl_2$ (3 × 50 mL). The combined organic layers were washed with water (3 × 100 mL), brine (100 mL) and dried over $MgSO_4$. The compound was purified by column chromatography (silica gel, eluting solvent 3:2 hexane: ethyl acetate) to yield a white solid of bisPFPA[1].H NMR (400 MHz, CDCl$_3$): $\delta \approx 4.68$ (s, 4H)[19];F-NMR (377 MHz, CDCl$_3$): $\delta \approx -150.64$, (m, 4F) and $-138.18$ (m, 4F); see Supplementary Fig. 2.

### Procedures for direct photolithography of MOF films

The UV light source was a 254 nm low-pressure mercury vapor grid lamp (Guangzhou Fusiao Special Lighting Instrument). In our setup, the exposure light intensity was about 3 mW cm$^{-2}$. Photomasks were designed by AutoCAD and composed of patterned chromium layers on quartz substrates. All procedures were performed under yellow light used for cleanroom lighting. The patterning included three steps: (1) Film coating: Colloidal MOFs were dispersed in

methanol or toluene with a concentration listed in Supplementary Table 2. The crosslinkers (bisPFPA) were dissolved in corresponding solvent (methanol or toluene) with a concentration of 5 mg mL$^{-1}$. Shortly before film coating, the solutions of colloidal MOFs and bisPFPA were mixed, where the mass ratio of bisPFPA to MOFs was 10–20 wt%. The fraction of bisPFPA can be lower for patterning purified MOF NPs with fewer ligands. For instance, thoroughly purified ZIF-8@BrijC10 NPs can be patterned with 1 wt% of bisPFPA (developing time, 20 min). The obtained ink was spin-coated on cleaned substrates (silicon, glass, quartz, indium tin oxide-coated glass, gold films on silicon, or flexible polyimide) at 2000 rpm for 30 s. This protocol yielded films with the thickness of about 200 nm for most of the MOFs used in this work. For HKUST-1@OA/OLAM, the obtained film thickness was about 70 nm. The thickness of spin-coated films can be tuned by changing the concentration of inks and/or spin-coating parameters. (2) UV exposure. Selected regions of the coated films were exposed to UV light (254 nm, 3 mW cm$^{-2}$) via pre-designed photomasks. Typical exposure doses were 90 mJ cm$^{-2}$. UV exposure can be performed either on a regular mask aligner or by placing the photomask atop films[81]. (3) Developing. The films were then immersed in the developer solvents to remove/redisperse MOF NPs in the unexposed regions. The developer was selected according to the polarity of the original solvents for MOF NPs. The time for developing also varied from 30 s to 30 min, mainly based on the colloidal stability of the pristine MOFs. After developing, the patterned films were dried under a gentle nitrogen flow. Detailed parameters for the direct photolithography of MOFs appear in Supplementary Table 2. To check the endurance of patterned MOF films on substrates, we performed the soaking tests by soaking the patterned films in different solvents (acetone, ethanol, toluene, methanol) for 24 h and checked the optical images afterwards.

## Procedures for direct e-beam lithography of MOF films

The film coating of HKUST-1@OA/OLAM used the same procedures described in the direct photolithography. E-beam writing was performed on a ThermoFisher Helios G4 UC Focused Ion Dual Beam microscope, at 5 kV acceleration voltage with 17.9 pA probe current. All patterns were written at 1 μs dwell time and 2.2 nm pitch size, while the pass (scan) numbers were adjusted in each exposure to obtain desired e-beam doses. Typical e-beam doses were 80 μC cm$^{-2}$. After the e-beam writing, the films were developed by immersing in toluene for 30 s to form nanoscale patterns.

## Procedures for Gas Sensing

Vapor sensing was performed by illuminating MOF patterns on quartz substrates, fixed positioned in a sealed chamber connected to a vapor generating system (Supplementary Fig. 23). A 635 nm laser (Thorlabs, red laser module, LDM635) was used as the light source. The light intensities of the first-order diffraction spots ($I_1$) were recorded in a real-time manner with light intensity meters (Thorlabs, USB Power Meter PM16−120, 400 − 1100 nm). The diffraction grating structures were patterned with ZIF-8@BrijC10 MOF NPs (with ≈10 wt%) of bisazide crosslinkers in the coating solution.

Upon the adsorption of analyte gases, the light passing through a ZIF pattern will generate a phase difference ($\varphi$),

$$\varphi = \frac{2\pi h(n_{ZIF} - n_{N2})}{\lambda} \quad (1)$$

where $\varphi$ is related to film thickness $h$, light wavelength $\lambda$, and refractive indices of the ZIF and the surrounding atmosphere, $n_{ZIF}$ and $n_{N2}$.

The adsorption of gas molecules increased the refractive index of ZIF-8@BrijC10 ($n_{ZIF}$), leading to larger $\varphi$. Thus, the analyte adsorption could be detected by monitoring the intensity changes of the first-

order diffraction spot ($I_1$), which is related to $\varphi$ as follows.

$$I_1 = \frac{2(1 - \cos\varphi)}{\pi^2} \quad (2)$$

The relative pressure ($P/P_0$) of gas analytes was thus connected to the changes in $I_1$, based on Eqs. (1) and (2). To check the durability of diffraction grating sensor in multi-cycle utilization, the grating was immersed in dichloromethane solvent for 8 h prior to testing, and tested in 80% vapor pressure ($P/P_0$) of ethanol for 3 times. This completed one cycle of solvent immersion/gas vapor sensing. The grating was then immersed in dichloromethane for another 8 h before the vapor response test, for the second and third cycle.

## Fabrication of the pixelated electrochromic Devices

A colloidal ZIF-8@BrijC10 NP solution (50 mg mL$^{-1}$) in methanol was mixed with NDI molecules (10 mg mL$^{-1}$) and vortexed for 2 h. Unbound NDI molecules were separated from MOF NPs by centrifuging the solution. The precipitates of MOF NPs were collected and redispersed in chloroform. This MOF NP solution was mixed with bisPFPA cross-linkers for patterning via CLIP-MOF (≈10 wt% to the mass of MOF NPs). The patterns were formed on an ITO-coated glass substrate and covered by another piece of ITO-coated glass. The device was filled with a small volume of electrolyte (0.1 M [(nBu)$_4$N]PF$_6$ in DMF) and encapsulated by using tapes. The color changes in the pixelated electrochromic device were driven by a Keithley 2400 Dual-Channel System Digital Power Meter. Cyclic voltammetry measurements of the MOF films with NDI molecules were carried out in a DMF solution containing 0.1 M [(nBu)$_4$N]PF$_6$ using three-electrode electrochemical cells (CHI1100, Shanghai Huachen) with the ITO substrate as the working electrode, platinum mesh as the counter electrode, and Glymercury electrode as the pseudo-reference electrode (100 mV s$^{-1}$ scan rate).

## Characterization techniques

TEM images of colloidal MOF NPs were captured using a JEOL JEM-2100F microscope. The SAED and HRTEM of MOFs were captured using a JEOL JEM-2010 microscope. For the SAED and HRTEM analysis, we spin-coated the solution containing ZIF-8@CTAB NPs and bisazide crosslinkers on a TEM grid. The formed MOF film was then exposed to UV light and developed to remove unreacted crosslinkers. These procedures replicated those used in CLIP-MOF patterning. During the SAED measurements, we used a low electron dose to minimize the damage to the MOF structures during data collection. SEM measurements of pristine and patterned MOF films were carried out on Hitachi SU-08010 microscope at 10 kV. EDS analysis was performed on Gemini SEM 500. DLS analysis of pristine and crosslinked MOF NPs were performed on a Malvern Nano-ZS Zetasizer. Optical microscopic images of MOF patterns in bright mode were taken with a Nikon Ni-U microscope. The PL emission spectra of Eu(BTC) films were collected on a Horiba FluoroMax Plus spectrometer. FTIR spectra of films were collected using a Bruker VERTEX 70 spectrometer in a transmission mode. The samples for FTIR measurements were prepared by spin-coating solutions of MOF NPs (with or without bisPFPA) on a KBr substrate and exposed to different UV doses to monitor the photolysis of bisPFPAs. XPS spectra were obtained on a ThermoFisher Scientific ESCALAB Xi$^+$ spectrometer using a monochromatic Al Kα source. The contrast curve or film retention was measured via inductively coupled plasma-optical emission spectroscopy (ICP-OES) analysis of Zn. AFM images and 3D topological data were obtained in the ScanAsyst mode by using a Bruker Dimension Icon scanning probe microscope. Height profiles of MOF patterns were collected on Bruker Dektak XT instrument. XRD patterns of MOF NPs and patterned MOF films were collected by using a Rigaku Smartlab diffractometer with Cu Kα radiation. The surface area of the MOF powders (pristine or treated with procedures in CLIP-MOF) was measured by the Brunauer-Emmett-Teller

(BET) method using nitrogen adsorption and desorption isotherms on a Micrometrics ASAP 2020 system. For MOF NPs treated with CLIP-MOF procedures, the MOF solution containing bisPFPA crosslinkers (up to 20 wt% to the mass of MOF NPs) were exposed to UV for extended period ($\approx$5 min, corresponding to 900 mJ cm$^{-2}$) to ensure complete crosslinking. The crosslinked MOF NPs were collected by centrifugation and washed with methanol three times to remove residual bisPFPA. The dried MOF NP powders were then used for BET analysis[1].H- and $^{19}$F-NMR spectra of bisPFPA were collected by a JEOL ECS-400 at 400 MHz.

## Reporting summary

Further information on research design is available in the Nature Portfolio Reporting Summary linked to this article.

## Data availability

The data that support the findings of this study are available from the corresponding authors upon request. Source data are provided with this paper.

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

## Acknowledgements

This work was financially supported by the National Key Research and Development Program of China (No. 2022YEA1206101 (H.Z.) and No. 2021YFB3200800 (M.T.)), National Natural Science Foundation of China (No. 21974079 and No. 22274087 (H.Z.), No. 22201289 (M.T.)), Tsinghua University Initiative Scientific Research Program (H.Z.), Tsinghua University Dushi Program (H.Z.), the Shanghai Pujiang Program (No. 21PJ1415200) (M.T.), the Strategic Priority Research Program of Chinese Academy of Sciences (No. XDB36000000) (J.L.), and New Cornerstone Investigator program (J.L.).

## Author contributions

H.Z. conceived the concept of this work. H.Z. and X.T. designed the experimental work, analyzed the data, and led in writing the manuscript. X.T. led the experimental work with support from F.L., Z.T., S.W., K.W., D.L., S.L., W.Liu., Z.F., W.Li. and H.Q. H.Z., J.L. and M.T. supervised the project. The manuscript was written through contributions of all authors. All authors have given approval to the final version of the manuscript.

## Competing interests

Authors declare no competing interests.
