## [Peer Review File · Nature Communications]

Crosslinking-induced patterning of MOFs by direct photo- and electron-beam lithographyREVIEWER COMMENTS

Reviewer #1 (Remarks to the Author):

After reading carefully the article, I recommend the rejection of this article for Nature Communications. This paper is very hard to read and to follow. Importantly claims are not backed up with data. For example they claim to have used five different MOFs: ZIF-8, ZIF-7, HKUST-1, UiO-66 and Eu(BTC) but where is the characterization of those MOFs. They are already general guidelines how to characterize a MOF (ACS Cent. Sci. 2020, 6, 8, 1255–1273) and for this work for ALL samples have to prove the crystal structure with PXRD in order to show they have MOF structure after lithography. To be honest I doubt it, as those structures are not that stable under the used conditions. In addition, the SI is very hard to follow and has to be logic restructured.

Reviewer #2 (Remarks to the Author):

The manuscript reports the patterning of MOFs by photo- and ebeam lithography to fabricate planar functional elements for gas sorbtion, sensing, and optics.

Although the concept proposed is not completely new, the authors have successfully applied it to MOFs, which makes their work unique.

However, there are 3 issues that need to be clarified in detail whether or not the manuscript is published:

1. MOF integrity. I consider it insufficient to demonstrate only 1 PXRD pattern for Zif-8 after UV exposure, which is already one of the most stable and rigid MOF. Does amorphization occur for each MOF? Is there a detailed analysis of the PXRD data? Deep UV irradiation from 30 sec to 30 min for each MOF can cause at least partial destruction of relatively weak coordination bond. How was it analyzed for all MOFs? Adressing these questions allows one to call new method non-destructive in Fig. 1d!
2. Endurance. The Authors explain in detail the principle and mechanism of the patterning, but how we can play with the adhesion of MOFs to different substrates? Are they held by van der Waals interactions? If so, then there are doubts about the endurance, fragility of the resulting pattern on the substrate, and multi-cycle utilization of the patterns. This can already be a serious disadvantage compared to other patterning methods of MOFs (described in Table S1).
3. Geometry. Despite the novelty of the concept, I strongly recommend the Authors to indicate the existing (tens of nm) record values for MOF patterning (in the introduction), regardless of the method used and its complexity (Table S1). It is dangerous enough to mislead readers about achievements in this field, even if the information is correctly presented in the supplementary. The second point is the use of nanoMOFs as precursors for the pattern. Here I see 2 challenges: nanoMOFs possess a lower degree of porosity and a higher concentration of defects, which will negatively affect their sorption properties compared to an "ideal" large crystals; also, as the Authors found, the roughness of the patterns is related to the size of the nanoMOFs, and the smaller the MOF, the lower the roughness, while the more defects can be detected. I would like to see some discussion about this in the manuscript.

Reviewer #3 (Remarks to the Author):

I have read and considered the paper by Tian et al. submitted for publication.

The authors report on the use of a ligand cross-linking chemistry combined with a variety of different metal organic frameworks (MOFs) that behaves as a negative-tone photo- and electron-beam lithography resist. Importantly they show that, at least for some cases, the MOFs maintain their functional properties through a sensing/electrochromic experiment.

The paper is well structured, the many experimental results are solid and the discussion always appropriate. A few questions/remarks:

Can you specify the amount of cross-linker used in the solutions of Fig. 5? This is particularly relevant as these patterns were tested to verify the MOFs functionality.

Did you perform any contrast curve at least of one of the resist formulations? I think it would add value to the paper.

Can you comment on the aging time of the resist? I mean, for how many days/weeks the solutions are stable and contrast curves (or processing parameters) are comparable?

Response to Reviews

Reviewer #1 (Remarks to the Author):

Comment: *After reading carefully the article, I recommend the rejection of this article for Nature Communications. This paper is very hard to read and to follow. Importantly claims are not backed up with data. For example, they claim to have used five different MOFs: ZIF-8, ZIF-7, HKUST-1, UiO-66 and Eu(BTC) but where is the characterization of those MOFs. There are already general guidelines how to characterize a MOF (ACS Cent. Sci. 2020, 6, 8, 1255–1273) and for this work for ALL samples have to prove the crystal structure with PXRD in order to show they have MOF structure after lithography. To be honest I doubt it, as those structures are not that stable under the used conditions. In addition, the SI is very hard to follow and has to be logic restructured.*

Our Response: We appreciate the reviewer for these comments. In this work, we developed direct photo- and ebeam lithography for various MOFs without using conventional resists or etchants. This patterning approach (termed as CLIP-MOF) exploits the recent developments of colloidal MOF nanoparticles (NPs) and the irradiation-triggered cross-linking chemistry between ligands on MOF NPs. Importantly, the mild patterning condition allows the patterned MOFs to preserve their initial structures, porosity, and other properties. In the original manuscript, we showed that ZIF-8 NPs preserved their crystal structures (Fig. 5a) and porous properties (Fig. 5b and Supplementary Fig. 13). BET surface area for pristine and patterned ZIF-8@CTAB were both $\sim 1700 \text{ cm}^2 \text{ g}^{-1}$. Consequently, patterned ZIF-8 diffractive gratings showed good sensitivity to gas molecules (Fig. 5d, e and Supplementary Fig. 14).

In the revised manuscript, as suggested by the reviewer, we performed detailed characterizations on the properties of various MOFs, including ZIF-8@BrijC10, ZIF-8@CTAB, ZIF-7@PEI, UiO-66@PAA, HKUST-1@OA, HKSUT-1@OA/OLAM and Eu(BTC)@PVP, before and after patterning. Following reported guidelines (ACS Cent. Sci. 2020, 6, 8, 1255–1273), we used powder-XRD, HRTEM/SEM, BET and FTIR data to support the claim that the patterned MOFs remained stable. The patterned MOFs preserved their crystal structures, morphology, porosity, and chemical integrity. These major changes have been included in Fig. 5 and the section of “Properties of patterned MOFs” in the revised manuscript. In addition, we also made relevant changes in the manuscript and restructured the Supporting Information for clarity.

Detailed responses and changes are as follows.

Comment #1: *For example, they claim to have used five different MOFs: ZIF-8, ZIF-7, HKUST-1, UiO-66 and Eu(BTC) but where is the characterization of those MOFs. There are already general guidelines how to characterize a MOF (ACS Cent. Sci. 2020,*

6, 8, 1255–1273) and for this work for ALL samples have to prove the crystal structure with PXRD in order to show they have MOF structure after lithography. To be honest I doubt it, as those structures are not that stable under the used conditions.

Our Response: We thank the reviewer for this very helpful comment. As suggested in previous references (e.g., *ACS Cent. Sci.* 2020, 6, 8, 1255–1273), key indicators to support the stability of MOFs after certain treatments include the retention of their crystallinity and porosity. We thus used the retention of crystallinity and porosity to validate the stability of patterned MOFs (or replica treated by procedures involved in CLIP-MOF, including the addition of cross-linkers, UV exposure, and solvent development).

As suggested by the reviewer, we compared the powder-XRD patterns of all MOFs we used before and after patterning to support the retention of their structures and their stability. Moreover, TEM, SEM, BET, and FTIR analysis were performed to show that patterned MOFs retain their structures, morphology, crystallinity, porosity, and chemical linking in the reticular system. The combined results support our claim that the CLIP-MOF is nondestructive to the intrinsic properties of MOFs.

1) The patterning of different MOFs in this work. In the original manuscript, we showed in Fig. 4 and Supplementary Fig. 12 that MOFs of ZIF-8@BrijC10, ZIF-8@CTAB, ZIF-7@PEI, UiO-66@PAA and HKUST-1@OA/OLAM, can be patterned with microscale resolution in a single-layered or multi-layered fashion. The EDS data revealed the different compositions of these MOFs, proving these MOFs can be readily patterned.

Figure R1. XRD patterns of pristine and CLIP-MOF patterned films of various MOFs. (a) ZIF-8@CTAB, (b) ZIF-7@PEI, (c) UiO-66@PAA, (d) HKUST-1@OA, (e) HKUAT-1@OA/OLAM, (f) Eu(BTC)@PVP. Simulated data (the red vertical lines at the bottom) is shown for reference. Note that the XRD patterns of pristine and patterned ZIF-8@BrijC10 MOFs were shown as Fig. 5a in the original manuscript (also Fig. 5a in the revised manuscript) and thus not included here.

2) The preservation of MOFs' crystal structures and crystallinity. We measured the powder XRD patterns of spin-coated films of various MOFs before (termed as pristine) and after patterning procedures in CLIP-MOF (termed as CLIP-MOF). These MOF films include ZIF-8@BrijC10 (this data has been shown in the original manuscript as Fig. 5a and thus not included in Fig. R1), ZIF-8@CTAB, ZIF-7@PEI, UiO-66@PAA, HKUST-1@OA, HKUST-1@OA/OLAM, and Eu(BTC)@PVP. As shown in **Fig. R1**, both the pristine and CLIP-MOF patterned films show similar diffraction patterns that match the simulated data. No impurity peaks, additional peak widening, or changes in the relative peak intensities were observed after patterning. Note that the relatively broad peaks of HKUST-1@OA/OLAM are due to their small particle sizes (~ 3 nm, as estimated by Scherrer equation). These results suggest that MOFs preserve their crystal structures after patterning¹.

To validate the crystal structures of patterned MOFs, we further performed Rietveld refinement analysis on powder XRD data of CLIP-MOF treated ZIF-8@BrijC10, ZIF-7@PEI, and UiO-66@PAA. The measured data matches well with the calculated diffraction patterns (**Fig. R2**). For example, patterned ZIF-8@BrijC10 exhibits a cubic phase with the I-43m space group and lattice parameters $a = b = c = 17.0886(55)$ Å, $\alpha = \beta = \gamma = 90^\circ$. These parameters are consistent with those reported for ZIF-8 (see ref.²). Similarly, patterned ZIF-7@PEI exhibits a hexagonal phase with the R-3 space group and lattice parameters $a = b = 23.0779(31)$ Å, $c = 15.6520(40)$ Å, $\alpha = \beta = 90^\circ$, $\gamma = 120^\circ$. The patterned UiO-66@PAA sample demonstrates a cubic phase with Fm-3m space group with $a = b = c = 20.7475(61)$ Å, $\alpha = \beta = \gamma = 90^\circ$ ³.

Figure R2. Rietveld refinement of XRD data for patterned MOF films. (a) ZIF-8@BrijC10, **(b)** ZIF-7@PEI, **(c)** UiO-66@PAA. The measured and simulated data are shown in red lines and “+” dots with their differences plotted in purple lines. The blue vertical bars indicate the allowed peak positions. Fitted lattice parameters are also included. Goodness of data fitting, (a) $R_p = 2.80\%$ and $R_{wp} = 3.26\%$, (b) $R_p = 4.87\%$ and $R_{wp} = 3.79\%$, (c) $R_p = 2.02\%$ and $R_{wp} = 5.58\%$.

Complementary to the XRD data analysis, the selected area electron diffraction (SAED) from high resolution TEM analysis provides microscopic information of the crystallinity of MOFs. For this experiment, we spin-coated the solution containing ZIF-8@CTAB NPs and bisazide cross-linkers on a TEM grid. The formed MOF film was then exposed to UV light and developed to remove unreacted cross-linkers. These

procedures replicated those used in CLIP-MOF patterning. During the SAED measurements, we used a low electron dose and short acquisition time to minimize the damage to the MOF structures. **Figure R3a** shows evident and sharp diffraction spots for both pristine and patterned MOF samples, indicating the preservation of their crystallinity at microscale. We also captured the HRTEM and corresponding EDS data of these samples. MOF NPs in both cases show well-defined shapes and homogeneous distribution of Zn (from the metal ions) and N (from the organic linkers). Fluorine only exists in the patterned sample due to the cross-linking reaction.

Figure R3. SAED patterns and EDS data of pristine and CLIP-MOF treated ZIF-8@CTAB films. (a) SAED patterns along the $[1-10]$ zone axis. **(b)** HRTEM images and EDS mapping. Scale bars, (a) 2 nm^{-1} , (b) 500 nm .

The collection of powder XRD data of pristine and patterned MOFs, Rietveld refinement analysis, and the SAED data supports that MOFs remain structurally stable during patterning.

3) The retention of porosity by BET analysis. In the original manuscript, we showed that ZIF-8@CTAB NP powders treated by CLIP-MOF procedures almost fully maintained their BET surface area (1660.3 versus $1721.3 \text{ cm}^2 \text{ g}^{-1}$). During revision, we measured and compared the porosity of other MOFs, including ZIF-8@BrijC10, UiO-66@PAA, HKUST-1@OA before and after patterning by BET analysis with nitrogen. ZIF-7@PEI was not tested because that nitrogen molecules are reportedly inaccessible to the small pores of ZIF-7 and instead only absorbed on the external surface⁴. For BET measurements, powders of MOFs were used directly (“pristine” sample) or after UV exposure with cross-linkers and solvent development (“CLIP-MOF” treated sample). The N_2 adsorption–desorption isotherms, BET surface area and pore size distribution of pristine and CLIP-MOF treated MOF powders are shown in **Fig. R4** and **Table R1**. The CLIP-MOF treated powders show similar BET surface area (reduction below $\sim 10\%$) and pore size distribution with the pristine ones; the values are also consistent

with those for corresponding MOF NPs reported in previous work (see Table R1).

Figure R4. N₂ adsorption–desorption isotherms and BJH size distribution of pristine and CLIP-MOF powders. (a–d) N₂ adsorption–desorption isotherms of (a) ZIF-8@BrijC10, (b) ZIF-8@CTAB, (c) UiO-66@PAA and (d) HKUST-1@OA. (e–h) Size distribution of (e) ZIF-8@BrijC10, (f) ZIF-8@CTAB, (g) UiO-66@PAA and (h) HKUST-1@OA.

Table R1. Summary of BET surface area, pore volume, and pore size of various MOF NP powders before and after CLIP-MOF treatment. *

MOFs	BET surface area (cm ² g ⁻¹)	total pore volume (cm ³ g ⁻¹)	micropore volume (cm ³ g ⁻¹)	pore diameter (nm)
ZIF-8@BrijC10	999.5	0.95	0.44	0.58
ZIF-8@BrijC10 CLIP-MOF	887.7	0.82	0.38	0.57
data in ref ⁵	1478.5	–	0.58	–
ZIF-8@CTAB	1721.3	1.22	0.63	0.87
ZIF-8@CTAB CLIP-MOF	1660.3	1.16	0.61	0.87
data in ref ⁶	1200	–	0.70	–
UiO-66@PAA	1209	1.13	0.49	0.57
UiO-66@PAA CLIP-MOF	1066	0.96	0.44	0.56
data in ref ⁷	1050	–	–	–

HKUST-1@OA	1400	1.11	0.57	0.43
HKUST-1@OA CLIP-MOF	1479	1.32	0.60	0.47
data in ref ⁸	1472			0.78

* Data in refs. show porosity data for MOF NPs synthesized in the same or similar approach and with similar sizes and ligands in previous reports.

4) Preserved coordination bonding between metal ions and organic linkers in MOFs before and after patterning. In addition to the preserved crystallinity and porosity, we also confirmed that the coordination bonding between metal ions and organic linkers in MOFs remain intact after CLIP-MOF patterning. In brief, we compared the FTIR spectra of pristine MOFs films, MOF films containing bisazide cross-linkers before UV exposure, and MOF films treated by CLIP-MOF procedures (UV exposure and development) (**Fig. R5**). The pristine samples show resonance related to C–H bonds (at 2800–3000 cm^{-1}) in the organic linkers as well as bonds between metal ions and the organic linkers. For instance, pristine ZIF-8 or ZIF-7 MOFs show strong Zn–N band at 421 cm^{-1} , as reported in ref.¹. The addition of bisazide based cross-linkers to pristine MOFs introduces additional peaks corresponding to the azido group (at $\sim 2100 \text{ cm}^{-1}$). For samples treated with CLIP-MOF procedures, resonance from the azido group disappeared while that for Zn–N band remain unchanged. Meanwhile, no additional peaks are observed at 1843 and 791 cm^{-1} for ZIFs, which excludes the possible degradation process involving N–H groups¹. This suggests that the patterning procedures do not damage the bonding in ZIF-8 or ZIF-7 MOFs. Similarly, the preservation of Cu–O band at 492 cm^{-1} and Zr–O band at 663 cm^{-1} for HKUST-1⁸ and UiO-66⁹, respectively, confirms the retention of bonding and structures of these patterned MOFs.

Figure R5. FTIR Spectra of MOF films during CLIP-MOF processes. (a) ZIF-8@CTAB, (b) ZIF-7@PEI, (c) UiO-66@PAA, (d) HKUST-1@OA/OLAM. In each case, the spectra of pristine films, films coated with bisazide-based cross-linkers, and

films treated with CLIP-MOF procedures (addition of cross-linkers, UV exposure, and solvent developing) are compared.

5) Preserved morphology of MOF NPs. We also checked the SEM images of patterned MOF films to rule out the possibility of structural damage of MOF NPs during patterning. To avoid structural degradation from the exposure to electron beam during SEM imaging, we used low electron acceleration voltages. As shown in **Fig. R6**, MOFs of various compositions, sizes, and surface ligands maintain their well-defined and faceted structures after patterning.

Figure R6. SEM images of various MOF films patterned by CLIP-MOF. (a) ZIF-8@CTAB, **(b)** ZIF-7@PEI, **(c)** UiO-66@PAA, **(d)** HKUST-1@OA, **(e)** HKUST-1@OA/OLAM, **(f)** Eu(BTC)@PVP. The insets in each panel show the magnified view, highlighting the preservation of faceted structures of MOF NPs. Scale bars, 1 μ m and 100 nm (insets).

Figure R7. Preserved crystallinity and porosity of patterned MOF films. (a–e) Powder XRD data of CLIP-MOF patterned films of (a) ZIF-8@BrijC10, (b) ZIF-8@CTAB, (c) ZIF-7@PEI, (d) UiO-66@PAA, (e) HKUST-1@OA. For comparison, XRD patterns of pristine MOF films and the standard diffraction patterns (indicated by vertical lines at the bottom) are included. (f–h) Rietveld refinement of powder XRD data for CLIP-MOF treated (f) ZIF-8@BrijC10, (g) ZIF-7@PEI, and (h) UiO-66@PAA powders. The measured and calculated data are shown in red lines and “+” dots, with their differences plotted in purple lines. The blue vertical bars indicate the allowed peak positions. Fitted lattice parameters are also included. Goodness of data fitting, (f) $R_p = 2.8\%$ and $R_{wp} = 3.26\%$, (g) $R_p = 4.87\%$ and $R_{wp} = 3.79\%$, (h) $R_p = 2.02\%$ and $R_{wp} = 5.58\%$. (i, j) SAED patterns of pristine and CLIP-MOF treated ZIF-8@CTAB films. SAED patterns along the $[1-10]$ zone axis. Scale bars, 2 nm^{-1} . (k–n) N_2 adsorption–desorption isotherms of various pristine and CLIP-MOF treated MOF NP powders. (k) ZIF-8@BrijC10, (l) ZIF-8@CTAB, (m) UiO-66@PAA, (n) HKUST-1@OA. (o) PL spectra of pristine and patterned Eu(BTC)@PVP films. Inset is the photograph of patterned Eu(BTC)@PVP, taken under UV light. The source image for making the pattern in o is obtained from *Visual China* website with permission.

Our Changes:

1) We added the powder XRD data (Fig. R1), Rietveld refinement analysis (Fig. R2), SAED pattern (Fig. R3a), and BET analysis (Fig. R4a–d) of pristine and patterned MOFs as **Fig. 5** (see **Fig. R7**) in the revised manuscript (Page 18).

Figure R7 (or Fig. 5 in the revised manuscript). **Preserved crystallinity and porosity of patterned MOF films.** **a–e** Powder XRD data of CLIP-MOF patterned films of a ZIF-8@BrijC10, b ZIF-8@CTAB, c ZIF-7@PEI, d UiO-66@PAA, e HKUST-1@OA. For comparison, XRD patterns of pristine MOF films and the standard diffraction patterns (indicated by vertical lines at the bottom) are included. **f–h** Rietveld refinement of powder XRD data for CLIP-MOF treated f ZIF-8@BrijC10, g ZIF-7@PEI, and h UiO-66@PAA powders. The measured and calculated data are shown in red lines and “+” dots, with their differences plotted in purple lines. The blue vertical bars indicate the allowed peak positions. Fitted lattice parameters are also included. Goodness of data fitting, f $R_p = 2.8\%$ and $R_{wp} = 3.26\%$, g $R_p = 4.87\%$ and $R_{wp} = 3.79\%$, h $R_p = 2.02\%$ and $R_{wp} = 5.58\%$. **i, j** SAED patterns of pristine and CLIP-MOF treated ZIF-8@CTAB films. SAED patterns along the [1–10] zone axis. Scale bars, 2 nm^{-1} . **k–n** N_2 adsorption–desorption isotherms of various pristine and CLIP-MOF treated MOF NP powders. k ZIF-8@BrijC10, l ZIF-8@CTAB, m UiO-66@PAA, n HKUST-1@OA. **o** PL spectra of pristine and patterned Eu(BTC)@PVP films. Inset is the photograph of patterned Eu(BTC)@PVP, taken under UV light. The source image for making the pattern in **o** is obtained from *Visual China* website with permission.

2) We added powder XRD data of pristine and patterned HKUST-1@OA/OLAM and Eu(BTC)@PVP as Supplementary Fig. 18 in the revised Supplementary Information.

3) We added the HRTEM images and EDS data of pristine and CLIP-MOF treated ZIF-8@CTAB films (Fig. R3b) as Supplementary Fig. 20 in the revised Supplementary Information.

4) We added the BJH size distribution (Fig. R4e–h) and the summary of BET analysis data (Table R1) of various pristine and CLIP-MOF powders as Supplementary Fig. 21 and Supplementary Table 3.

5) We added the FTIR data showing the preservation of metal ion–organic linker bonds in patterned MOFs (Fig. R5) as Supplementary Fig. 22 in the revised Supplementary Information.

6) We added paragraphs in the section of “Properties of patterned MOFs” in the revised manuscript to discuss the above data (Page 19–21).

“CLIP-MOF is nondestructive to the intrinsic properties of MOFs, including the crystallinity, structures, porosity, and coordination bonding, as supported by data from

a collection of characterization techniques. As shown in the powder XRD data in Fig. 5a–e and Supplementary Fig. 18, MOF NPs including ZIF-8@BrijC10, ZIF-8@CTAB, ZIF-7@PEI, UiO-66@PAA, HKUST-1@OA and HKUST-1@OA/OLAM, and Eu(BTC)@PVP maintain their crystal structure and crystallinity after being treated by the CLIP-MOF patterning procedures (addition of cross-linkers with the amount of 5–20 wt% to the mass of MOFs, UV exposure, and removal of unreacted cross-linkers by developer solvents). No impurity peaks, additional peak widening, or changes in the relative peak intensities were observed after patterning. Owing to the high efficiency in the cross-linking chemistry and low UV doses required for patterning, the XRD patterns of patterned MOF films remain identical during UV exposure (up to 1.8 J cm^{-2} , or 20 times higher than the required dose for patterning), as shown in Supplementary Fig. 19. To validate the crystal structures of patterned MOFs, we further performed Rietveld refinement analysis on XRD data of patterned ZIF-8@BrijC10, ZIF-7@PEI and UiO-66@PAA films, where the measured data matches well with the calculated diffraction patterns (Fig. 5f–h). In brief, patterned ZIF-8@BrijC10 exhibits a cubic phase with the $I-43m$ space group and lattice parameters $a = b = c = 17.0886(55) \text{ \AA}$, $\alpha = \beta = \gamma = 90^\circ$. These parameters are consistent with those reported for ZIF-8⁵⁹. Similarly, patterned ZIF-7@PEI and UiO-66@PAA samples exhibit a hexagonal phase ($R-3$ space group)⁵⁹ and a cubic phase ($Fm-3m$ space group)⁶⁰, respectively, consistent with reported phases. Complementary to the XRD data analysis, the selected area electron diffraction (SAED) patterns from high resolution TEM (HRTEM) analysis provide microscopic information of the crystallinity of MOFs. Figure 5i, j shows evident and sharp diffraction spots for both pristine and patterned ZIF-8@CTAB samples, indicating the preservation of their crystallinity at microscale. We also captured the HRTEM and corresponding EDS data of these samples (Supplementary Fig. 20). MOF NPs in both cases show well-defined shapes and homogeneous distribution of Zn (from the metal ions) and N elements (from the organic linkers). Fluorine only exists in the patterned sample due to the cross-linking reaction.

The unchanged framework results in almost identical nitrogen adsorption and desorption isotherms of pristine and treated samples (Fig. 5k–n and Supplementary Table 3). For instance, the estimated Brunauer-Emmett-Teller (BET) surface area of pristine and patterned ZIF-8@CTAB NP MOF are 1721.3 and $1660.3 \text{ cm}^2 \text{ g}^{-1}$, respectively. These values are on par with those reported for colloidal ZIF-8 MOFs ($900\text{--}1630 \text{ cm}^2 \text{ g}^{-1}$)⁶¹. The pristine and patterned MOFs also show similar pore size distribution (Supplementary Fig. 21). The fully preserved porous properties follow from the absence of resist contamination or plasma/etchant damages, which are the major concerns for conventional photolithography of MOFs³³.

In addition to the preserved crystallinity and porosity, we also confirmed that the bonding between metal ions and organic linkers in MOFs remain intact before and after patterning. Supplementary Fig. 22 compares the FTIR spectra of pristine MOFs films, MOF films containing bisazide cross-linkers before UV exposure, and MOF films treated by CLIP-MOF procedures. The pristine samples show resonance related to C–

H bonds (at 2800–3000 cm^{-1}) in the organic linkers as well as bonds between metal ions and the organic linkers. For instance, pristine ZIF-8 or ZIF-7 MOFs show strong Zn–N band at 421 cm^{-1} ⁶². The addition of bisazide based cross-linkers to pristine MOFs introduces additional peaks corresponding to the azido group (e.g., $\sim 2100 \text{ cm}^{-1}$). For ZIF MOF samples treated with CLIP-MOF procedures, resonance from the azido group disappeared while that for Zn–N band remain unchanged. Meanwhile, no additional peaks are observed at 1843 and 791 cm^{-1} for ZIFs, which excludes the potential degradation process involving the N–H groups⁶². Similarly, the preservation of Cu–O band at 492 cm^{-1} and Zr–O band at 663 cm^{-1} for HKUST-1⁶³ and UiO-66⁶⁴, respectively, confirms the retention of chemical bonding and structures of these patterned MOFs.”

7) We added procedures related to these HRTEM and SAED measurements in the revised “Methods” (Page 32–33).

“The SAED and HRTEM of MOFs were captured using a JEOL JEM-2010 microscope. For the SAED and HRTEM analysis, we spin-coated the solution containing ZIF-8@CTAB NPs and bisazide cross-linkers on a TEM grid. The formed MOF film was then exposed to UV light and developed to remove unreacted cross-linkers. These procedures replicated those used in CLIP-MOF patterning. During the SAED measurements, we used a low electron dose to minimize the damage to the MOF structures during data collection.”

8) We added the SEM images of patterned MOF films (Fig. R6) as Supplementary Fig. 15 in the revised Supplementary Information.

9) We added related descriptions for the SEM images of MOF films in the revised manuscript to support the claim that MOFs retain their structures after patterning (Page 15–16).

“The magnified SEM images of these patterned MOF films (Supplementary Fig. 15) also rule out the possibility of structural damage or amorphization of MOF NPs. MOFs of various compositions, sizes, and surface ligands maintain their well-defined and faceted structures.”

Comment #2: *This paper is very hard to read and to follow. In addition, the SI is very hard to follow and has to be logic restructured.*

Our Response: We thank the reviewer for these helpful comments. We made major changes to the manuscript and SI to include the changes and additional figures shown in the response to Comment #1 to support the preservation of properties of patterned MOFs. We also made additional changes in the manuscript and SI for clarity.

Our Changes: Some of the changes include the following.

1) We restructured the “Results” section of the revised manuscript. The updated manuscript contains “Patterning mechanism and chemistry”, “Direct photolithography of MOF films with microscale resolution”, “Properties of patterned MOFs”, “Example applications of patterned MOF films”, and “Direct ebeam lithography of MOF films with nanoscale resolution”. These sections follow the orders of underlying chemistry for patterning, patterning capabilities and material adaptabilities via direct photolithography, preservation of the intrinsic properties of patterned MOFs (crystallinity, porosity, coordination bonding, etc.), their applications as diffractive grating sensors and electrochromic devices, and also the possibility of nanoscale patterning with ebeam irradiation.

2) We added discussions in the revised manuscript for clarification or providing detailed characterizations of the method. Examples include,

Page 14, “We further studied the adhesion stability and film retention curve of patterned MOF films.” and the following paragraph for describing the endurance and film retention curves of patterned MOF NP films.

Pages 15, 16, “The magnified SEM images of these patterned MOF films (Supplementary Fig. 15) also rule out the possibility of structural damage or amorphization of MOF NPs during patterning. MOFs of various compositions, sizes, and surface ligands maintain their well-defined and faceted structures.” for supporting the claim that there is no structural damage or material deterioration during patterning.

Major changes have been made to the section of “Properties of patterned MOFs” and Fig. 5 to support the nondestructive nature of CLIP-MOF.

3) Associated with these changes in the revised manuscript, we also included additional Supplementary Figures (Figs. S12–S22 and S24) and Table (Table S3) in the revised SI. The restructured SI contains supplementary methods on the synthetic approaches of MOF NPs and cross-linkers, and supplementary figures and tables for the characterizations of these raw materials, additional information on the patterning chemistry, patterning capabilities, and properties and applications of patterned MOFs.

We hope the revised/restructured manuscript and SI can help clarify the design and features of CLIP-MOF.

Reviewer #2 (Remarks to the Author):

Comments: The manuscript reports the patterning of MOFs by photo- and ebeam lithography to fabricate planar functional elements for gas sorption, sensing, and optics. Although the concept proposed is not completely new, the authors have successfully applied it to MOFs, which makes their work unique.

Our Response: We thank the review for these positive comments. In the revision, we performed additional experiments to validate the unique features of CLIP-MOF, including the preservation of the MOF integrity and the endurance of obtained MOF patterns during subsequent and repeated treatments and applications. We also clarified the difference between our method and other MOF patterning methods as well as included the considerations of using colloidal MOFs as the precursors.

Comment #1: *MOF integrity. I consider it insufficient to demonstrate only 1 PXRD pattern for ZIF-8 after UV exposure, which is already one of the most stable and rigid MOF. Does amorphization occur for each MOF? Is there a detailed analysis of the PXRD data? Deep UV irradiation from 30 sec to 30 min for each MOF can cause at least partial destruction of relatively weak coordination bond. How was it analyzed for all MOFs? Addressing these questions allows one to call new method non-destructive in Fig. 1d!*

Our Response: We thank the reviewer for the helpful comments. We agree with the reviewer that detailed analysis of the structural and chemical stability of different MOFs after patterning is critical to support the claim of nondestructive patterning. In the revised manuscript, we used a combination of powder XRD data, Rietveld refinement, selected area electron diffraction (SAED) analysis in high-resolution TEM (HRTEM), and FTIR spectra to confirm the preservation of the crystallinity, structures, and bonding in various MOFs after patterning or treated with procedures in CLIP-MOF (namely, the addition of bisazide cross-linkers, UV exposure, and developing with solvents). The nondestructive nature comes from the low UV exposure doses ($<150 \text{ mJ cm}^{-2}$) required for patterning. To test this, we also monitored the powder XRD patterns of various MOFs during UV irradiation (254 nm) with doses up to $\sim 1.8 \text{ J cm}^{-2}$, which is almost 20 times higher than the required dose for patterning. The crystallinity of different MOFs was maintained.

1) Powder XRD data showing the preservation of crystal structures and crystallinity for different MOFs. As suggested by the reviewer, we measured the powder XRD data of spin-coated films of various MOFs before (termed as pristine) and after procedures in CLIP-MOF (termed as patterned or CLIP-MOF). Beside the ZIF-8@BrijC10 shown in the original manuscript, other MOFs, including ZIF-8@CTAB, ZIF-7@PEI, UiO-66@PAA, HKUST-1@OA, HKUST-1@OA/OLAM and Eu(BTC)@PVP, also preserve their crystal structures and crystallinity. As shown in **Fig. R8**, both the pristine and CLIP-MOF patterned MOF films show similar diffraction

patterns that match the simulated data. No impurity peaks, additional peak widening, or changes in the relative peak intensities were observed after patterning. Note that the relatively broad peaks of HKUST-1@OA/OLAM are due to their small particle sizes (~ 3 nm, as estimated by Scherrer equation). These results suggest that MOFs preserve their crystal structures after patterning without amorphization¹.

Figure R8. XRD patterns of pristine and CLIP-MOF patterned films of various MOFs. (a) ZIF-8@CTAB, (b) ZIF-7@PEI, (c) UiO-66@PAA, (d) HKUST-1@OA, (e) HKUAT-1@OA/OLAM, (f) Eu(BTC)@PVP. Simulated data (the red vertical lines at the bottom) is shown for reference. Note that the XRD patterns of pristine and patterned ZIF-8@BrijC10 MOFs were shown as Fig. 5a in the original manuscript (also Fig. 5a in the revised manuscript) and thus not included here.

2) Detailed analysis of the powder XRD data. To validate the crystal structures of patterned MOFs, we further performed Rietveld refinement analysis on powder XRD data of CLIP-MOF treated ZIF-8@BrijC10, ZIF-7@PEI and UiO-66@PAA MOF NP powders, where the measured data matches well with the calculated diffraction patterns (Fig. R9). In brief, patterned ZIF-8@BrijC10 exhibits a cubic phase with the I-43m space group and lattice parameters $a = b = c = 17.0816(13)$ Å, $\alpha = \beta = \gamma = 90^\circ$. These parameters are consistent with those reported for ZIF-8, as shown in ref.². Similarly, patterned ZIF-7@PEI exhibits a hexagonal phase with the R-3 space group and lattice parameters $a = b = 23.0779(31)$ Å, $c = 15.6520(40)$ Å, $\alpha = \beta = 90^\circ$, $\gamma = 120^\circ$. The patterned UiO-66@PAA sample demonstrates a cubic phase with Fm-3m space group with $a = b = c = 20.7475(61)$ Å, $\alpha = \beta = \gamma = 90^\circ$ ³.

Figure R9. Rietveld refinement of XRD data for patterned MOF films. (a) ZIF-8@BrijC10, (b) ZIF-7@PEI, (c) UiO-66@PAA. The measured and simulated data are shown in red lines and “+” dots with their differences plotted in purple lines. The blue vertical bars indicate the allowed peak positions. Fitted lattice parameters are also included. Goodness of data fitting, (a) $R_p = 2.80\%$ and $R_{wp} = 3.26\%$, (b) $R_p = 4.87\%$ and $R_{wp} = 3.79\%$, (c) $R_p = 2.02\%$ and $R_{wp} = 5.58\%$.

3) Microscopic imaging to rule out the possibility of amorphization.

Complementary to the XRD data analysis, the SAED from HRTEM analysis provides microscopic information of the crystallinity of MOFs. For this experiment, we spin-coated the solution containing ZIF-8@CTAB NPs and bisazide cross-linkers on a TEM grid. The formed MOF film was then exposed to UV light and developed to remove unreacted cross-linkers. These procedures replicated those used in CLIP-MOF patterning. During the SAED measurements, we used a low electron dose to minimize the damage to the MOF structures. **Figure R10a** shows evident and sharp diffraction spots for both pristine and patterned MOF samples, indicating the preservation of their crystallinity. We also captured the HRTEM and corresponding EDS data of these samples (Fig. R10b). MOF NPs in both cases show well-defined shapes and homogeneous distribution of Zn (from the metal ions) and N (from the organic linkers). Fluorine only exists in the patterned sample due to the cross-linking reaction.

Figure R10. SAED patterns and EDS data of pristine and CLIP-MOF treated ZIF-8@CTAB films. (a) SAED patterns along the $[1-10]$ zone axis. (b) HRTEM images and EDS mapping. Scale bars, (a) 2 nm^{-1} , (b) 500 nm .

Other than ZIF-8, we also checked the SEM images of patterned MOF films of different compositions to rule out the possibility of structural damage or amorphization. To avoid structural degradation from the exposure to electron beam during image capturing, we

used low electron acceleration voltage for imaging. As shown in **Fig. R11**, MOFs of various compositions, sizes, and surface ligands maintain their well-defined and faceted structures after patterning.

All the above data excludes the possibility of amorphization of MOFs during patterning and supports the nondestructive nature of the CLIP-MOF patterning.

Figure R11. SEM images of various MOF films patterned by CLIP-MOF. (a) ZIF-8@CTAB, **(b)** ZIF-7@PEI, **(c)** UiO-66@PAA, **(d)** HKUST-1@OA, **(e)** HKUST-1@OA/OLAM, **(f)** Eu(BTC)@PVP. The insets in each panel show the magnified view, highlighting the preservation of faceted structures of MOF NPs. Scale bars, 1 μm and 100 nm (insets).

4) The preservation of the coordination bonding between metal ions and organic linkers in MOFs before and after patterning. In addition to the preserved crystallinity and structures, we also confirmed that the coordination bonding between metal ions and organic linkers in MOFs remain intact before and after CLIP-MOF patterning. In brief, we compared the FTIR spectra of pristine MOFs films, MOF films containing bisazide cross-linkers before UV exposure, and MOF films treated by CLIP-MOF procedures (UV exposure and development) (**Fig. R12**). The pristine samples show resonance related to C–H bonds (at 2800–3000 cm^{-1}) in the organic linkers as well as bonds between metal ions and the organic linkers. For instance, pristine ZIF-8 or ZIF-7 MOFs show strong Zn–N band at 421 cm^{-1} (see ref.¹). The addition of bisazide based cross-linkers to pristine MOFs introduces additional peaks corresponding to the

azido group (e.g., at $\sim 2100\text{ cm}^{-1}$). For samples treated with CLIP-MOF procedures, resonance from the azido group disappeared while that for Zn–N band remained unchanged. This suggests that the patterning procedures do not damage the bonding in ZIF-8 or ZIF-7 MOFs. Meanwhile, no additional peaks are observed at 1843 and 791 cm^{-1} for ZIFs, which excludes the potential degradation involving N–H groups¹. Similarly, the preservation of Cu–O band at 492 cm^{-1} and Zr–O band at 663 cm^{-1} for HKUST-1⁸ and UiO-66⁹, respectively, confirms the retention of chemical bonding and structures of these patterned MOFs. These results suggest that the short deep UV irradiation for CLIP-MOF patterning does not cause destruction of the coordination bonds in MOFs.

Figure R12. FTIR Spectra of MOF films during CLIP-MOF processes. (a) ZIF-8@CTAB, (b) ZIF-7@PEI, (c) UiO-66@PAA, (d) HKUST-1@OA. In each case, the spectra of pristine films, films coated with bisazide-based cross-linkers, and films treated with CLIP-MOF procedures (addition of cross-linkers, UV exposure, and solvent developing) are compared.

5) XRD patterns of MOFs during long-term UV irradiation. CLIP-MOF represents an efficient patterning method that requires low UV doses of $\sim 100\text{ mJ cm}^{-2}$. To test the stability of various MOFs under deep UV irradiation (254 nm), we monitored the changes in the XRD patterns of both the pristine and patterned MOFs at increasing UV doses up to $\sim 1.8\text{ J cm}^{-2}$, which is about 20 times the required dose for patterning. As shown in **Fig. R13**, the XRD data of patterned MOF films, including ZIF-8@BrijC10, ZIF-7@PEI and UiO-66@PAA remain almost unchanged during UV exposure.

Previous work also investigated the stability of MOFs during prolonged UV exposure. For example, UV exposure of high doses (254 nm UV light irradiation $>30\text{ min}$, 650 J cm^{-2}) resulted in partial structural disconnection of ZIF-8 and an increase in the number of end-face species (e.g., Zn–OH and C=N–H)¹, although the overall structure, specific surface area or thermal stability of the MOFs can be preserved. In another report, HKUST-1 can endure UV irradiation for 30 min (900 J cm^{-2} , $500\text{--}2650\text{ mW cm}^{-2}$) without being destroyed¹⁰. In our case, CLIP-MOF requires much lower UV exposure

(< 150 mJ cm⁻², or 3 orders of magnitude lower than a few hundreds of J cm⁻² in previous reports) and brief irradiation time (< 90 s) and thus maintains the structures and coordination bonding of patterned MOFs.

Figure R13. XRD patterns of films of pristine and patterned MOFs collected after UV exposure with different doses. (a) ZIF-8@BrijC10, (b) ZIF-8@BrijC10 (CLIP-MOF), (c) ZIF-7@PEI, (d) ZIF-7@PEI (CLIP-MOF), (e) UiO-66@PAA, (f) UiO-66@PAA (CLIP-MOF). Note that regular patterning requires UV doses of ~90 mJ cm⁻². UV exposure does not cause notable changes in the crystal structures of MOFs.

Figure R14. Preserved crystallinity and porosity of patterned MOF films. (a–e) Powder XRD data of CLIP-MOF patterned films of (a) ZIF-8@BrijC10, (b) ZIF-8@CTAB, (c) ZIF-7@PEI, (d) UiO-66@PAA, (e) HKUST-1@OA. For comparison, XRD patterns of pristine MOF films and the standard diffraction patterns (indicated by vertical lines at the bottom) are included. (f–h) Rietveld refinement of powder XRD data for CLIP-MOF treated (f) ZIF-8@BrijC10, (g) ZIF-7@PEI, and (h) UiO-66@PAA powders. The measured and calculated data are shown in red lines and “+” dots, with their differences plotted in purple lines. The blue vertical bars indicate the allowed peak positions. Fitted lattice parameters are also included. Goodness of data fitting, (f) $R_p = 2.8\%$ and $R_{wp} = 3.26\%$, (g) $R_p = 4.87\%$ and $R_{wp} = 3.79\%$, (h) $R_p = 2.02\%$ and $R_{wp} = 5.58\%$. (i, j) SAED patterns of pristine and CLIP-MOF treated ZIF-8@CTAB films. SAED patterns along the $[1-10]$ zone axis. Scale bars, 2 nm^{-1} . (k–n) N_2 adsorption–desorption isotherms of various pristine and CLIP-MOF treated MOF NP powders. (k) ZIF-8@BrijC10, (l) ZIF-8@CTAB, (m) UiO-66@PAA, (n) HKUST-1@OA. (o) PL spectra of pristine and patterned Eu(BTC)@PVP films. Inset is the photograph of patterned Eu(BTC)@PVP, taken under UV light. The source image for making the pattern in o is obtained from *Visual China* website with permission.

Our Changes:

1) We added the powder XRD data (Fig. R8), Rietveld refinement analysis (Fig. R9), and SAED pattern (Fig. R10a) of patterned MOFs as **Fig. 5** (see **Fig. R14**) in the revised manuscript (Page 18).

Figure R14 (or Fig. 5 in the revised manuscript). **Preserved crystallinity and porosity of patterned MOF films.** **a–e** Powder XRD data of CLIP-MOF patterned films of a ZIF-8@BrijC10, b ZIF-8@CTAB, c ZIF-7@PEI, d UiO-66@PAA, e HKUST-1@OA. For comparison, XRD patterns of pristine MOF films and the standard diffraction patterns (indicated by vertical lines at the bottom) are included. **f–h** Rietveld refinement of powder XRD data for CLIP-MOF treated f ZIF-8@BrijC10, g ZIF-7@PEI, and h UiO-66@PAA powders. The measured and calculated data are shown in red lines and “+” dots, with their differences plotted in purple lines. The blue vertical bars indicate the allowed peak positions. Fitted lattice parameters are also included. Goodness of data fitting, f $R_p = 2.8\%$ and $R_{wp} = 3.26\%$, g $R_p = 4.87\%$ and $R_{wp} = 3.79\%$, h $R_p = 2.02\%$ and $R_{wp} = 5.58\%$. **i, j** SAED patterns of pristine and CLIP-MOF treated ZIF-8@CTAB films. SAED patterns along the [1–10] zone axis. Scale bars, 2 nm^{-1} . **k–n** N_2 adsorption–desorption isotherms of various pristine and CLIP-MOF treated MOF NP powders. k ZIF-8@BrijC10, l ZIF-8@CTAB, m UiO-66@PAA, n HKUST-1@OA. **o** PL spectra of pristine and patterned Eu(BTC)@PVP films. Inset is the photograph of patterned Eu(BTC)@PVP, taken under UV light. The source image for making the pattern in **o** is obtained from *Visual China* website with permission.

2) We added powder XRD data of pristine and patterned HKUST-1@OA/OLAM and Eu(BTC)@PVP as Supplementary Fig. 18 in the revised Supplementary Information.

3) We added the HRTEM images and EDS data of pristine and CLIP-MOF treated ZIF-8@CTAB films (Fig. R10) as Supplementary Fig. 20 in the revised Supplementary Information.

4) We added the FTIR data showing the preservation of metal ion–organic linker coordination bonds in patterned MOFs (Fig. R12) as Supplementary Fig. 22 in the revised Supplementary Information.

5) We added the XRD data of pristine and patterned MOF films during UV exposure (254 nm, up to 1.8 J cm^{-2} , Fig. R13) as Supplementary Fig. 19 in the revised Supplementary Information.

6) We added a paragraph in the section of “Properties of patterned MOFs” in the revised manuscript to discuss the above data (Page 19–21).

“CLIP-MOF is nondestructive to the intrinsic properties of MOFs, including the crystallinity, structures, porosity, and coordination bonding, as supported by data from

a collection of characterization techniques. As shown in the powder XRD data in Fig. 5a–e and Supplementary Fig. 18, MOF NPs including ZIF-8@BrijC10, ZIF-8@CTAB, ZIF-7@PEI, UiO-66@PAA, HKUST-1@OA and HKUST-1@OA/OLAM, and Eu(BTC)@PVP maintain their crystal structure and crystallinity after being treated by the CLIP-MOF patterning procedures (addition of cross-linkers with the amount of 5–20 wt% to the mass of MOFs, UV exposure, and removal of unreacted cross-linkers by developer solvents). No impurity peaks, additional peak widening, or changes in the relative peak intensities were observed after patterning. Owing to the high efficiency in the cross-linking chemistry and low UV doses required for patterning, the XRD patterns of patterned MOF films remain identical during UV exposure (up to 1.8 J cm^{-2} , or 20 times higher than the required dose for patterning), as shown in Supplementary Fig. 19. To validate the crystal structures of patterned MOFs, we further performed Rietveld refinement analysis on XRD data of patterned ZIF-8@BrijC10, ZIF-7@PEI and UiO-66@PAA films, where the measured data matches well with the calculated diffraction patterns (Fig. 5f–h). In brief, patterned ZIF-8@BrijC10 exhibits a cubic phase with the $I-43m$ space group and lattice parameters $a = b = c = 17.0886(55) \text{ \AA}$, $\alpha = \beta = \gamma = 90^\circ$. These parameters are consistent with those reported for ZIF-8⁵⁹. Similarly, patterned ZIF-7@PEI and UiO-66@PAA samples exhibit a hexagonal phase ($R-3$ space group)⁵⁹ and a cubic phase ($Fm-3m$ space group)⁶⁰, respectively, consistent with reported phases. Complementary to the XRD data analysis, the selected area electron diffraction (SAED) patterns from high resolution TEM (HRTEM) analysis provide microscopic information of the crystallinity of MOFs. Figure 5i, j shows evident and sharp diffraction spots for both pristine and patterned ZIF-8@CTAB samples, indicating the preservation of their crystallinity at microscale. We also captured the HRTEM and corresponding EDS data of these samples (Supplementary Fig. 20). MOF NPs in both cases show well-defined shapes and homogeneous distribution of Zn (from the metal ions) and N elements (from the organic linkers). Fluorine only exists in the patterned sample due to the cross-linking reaction.

The unchanged framework results in almost identical nitrogen adsorption and desorption isotherms of pristine and treated samples (Fig. 5k–n and Supplementary Table 3). For instance, the estimated Brunauer-Emmett-Teller (BET) surface area of pristine and patterned ZIF-8@CTAB NP MOF are 1721.3 and $1660.3 \text{ cm}^2 \text{ g}^{-1}$, respectively. These values are on par with those reported for colloidal ZIF-8 MOFs ($900\text{--}1630 \text{ cm}^2 \text{ g}^{-1}$)⁶¹. The pristine and patterned MOFs also show similar pore size distribution (Supplementary Fig. 21). The fully preserved porous properties follow from the absence of resist contamination or plasma/etchant damages, which are the major concerns for conventional photolithography of MOFs³³.

In addition to the preserved crystallinity and porosity, we also confirmed that the bonding between metal ions and organic linkers in MOFs remain intact before and after patterning. Supplementary Fig. 22 compares the FTIR spectra of pristine MOFs films, MOF films containing bisazide cross-linkers before UV exposure, and MOF films treated by CLIP-MOF procedures. The pristine samples show resonance related to C–

H bonds (at 2800–3000 cm^{-1}) in the organic linkers as well as bonds between metal ions and the organic linkers. For instance, pristine ZIF-8 or ZIF-7 MOFs show strong Zn–N band at 421 cm^{-1} ⁶². The addition of bisazide based cross-linkers to pristine MOFs introduces additional peaks corresponding to the azido group (e.g., ~2100 cm^{-1}). For ZIF MOF samples treated with CLIP-MOF procedures, resonance from the azido group disappeared while that for Zn–N band remain unchanged. Meanwhile, no additional peaks are observed at 1843 and 791 cm^{-1} for ZIFs, which excludes the potential degradation process involving the N–H groups⁶². Similarly, the preservation of Cu–O band at 492 cm^{-1} and Zr–O band at 663 cm^{-1} for HKUST-1⁶³ and UiO-66⁶⁴, respectively, confirms the retention of chemical bonding and structures of these patterned MOFs.”

7) We added procedures related to these HRTEM and SAED measurements in the revised “Methods” (Page 32–33).

“The SAED and HRTEM of MOFs were captured using a JEOL JEM-2010 microscope. For the SAED and HRTEM analysis, we spin-coated the solution containing ZIF-8@CTAB NPs and bisazide cross-linkers on a TEM grid. The formed MOF film was then exposed to UV light and developed to remove unreacted cross-linkers. These procedures replicated those used in CLIP-MOF patterning. During the SAED measurements, we used a low electron dose to minimize the damage to the MOF structures during data collection.”

8) We added the SEM images of patterned MOF films (Fig. R11) as Supplementary Fig. 15 in the revised Supplementary Information.

9) We added related descriptions for the SEM images of MOF films in the revised manuscript to support the claim that MOFs retain their structures after patterning (Page 15–16).

“The magnified SEM images of these patterned MOF films (Supplementary Fig. 15) also rule out the possibility of structural damage or amorphization of MOF NPs. MOFs of various compositions, sizes, and surface ligands maintain their well-defined and faceted structures.”

Comment #2: *Endurance. The Authors explain in detail the principle and mechanism of the patterning, but how we can play with the adhesion of MOFs to different substrates? Are they held by van der Waals interactions? If so, then there are doubts about the endurance, fragility of the resulting pattern on the substrate, and multi-cycle utilization of the patterns. This can already be a serious disadvantage compared to other patterning methods of MOFs (described in Table S1).*

Our Response: We thank the reviewer for this very helpful comment. In the original manuscript, we have shown that the patterning can be performed on different substrates, including silicon, glass, quartz, indium tin oxide-coated glass, gold films on silicon, and

flexible polyimide substrates (Supplementary Fig. 11). These substrates were not treated physically or chemically. The adhesion of MOF patterns on these substrates mostly stems from van der Waals interactions. According to ref.¹¹, the van der Waals forces between spherical nanoparticles (NPs) and a flat surface can be calculated by the equation $F = -AR/6D^2$, where A is the Hamaker constant of the materials of spheres, R is the radius of spheres, and D is the distance between spheres and the flat surface. The attractive forces between NP/surface are about twice of those between two identical NPs. For instance, for ZIF-8 NPs with the diameter of 30 nm, A is 3.45×10^{-20} J¹² and R is 15 nm. At a small separation between NPs and the substrate (e.g., D = 1 nm), the attractive van der Waals force is in the scale of a few nN, which is nontrivial for small NPs. In addition, after patterning, the R for cross-linked MOF NPs can be significantly increased to the feature size of the patterns (i.e., in microscale) and the NP/surface distance can be smaller than 1 nm. Therefore, the van der Waals forces can increase by more than 100 times and be adequate to hold the MOF films on substrates. We also noted that in recent reports of direct optical 2D patterning of colloidal inorganic nanocrystals or quantum dots, the patterned nanomaterials are held firmly to substrates without surface treatments via solely van der Waals forces^{13, 14, 15}. In those cases, the van der Waals forces between inorganic nanocrystals or quantum dots and the substrates are similar or even slightly smaller than those between MOFs and substrates. These observations also support the claim that the MOF patterns can stay firmly via van der Waals forces on substrates without treatments.

This assumption can be supported by our experimental results on the endurance of the MOF patterns (related to the adhesion of MOF layers on substrates) in several fabrication and utilization scenarios. These scenarios include **1)** multilayered patterning, **2)** the soaking test of patterned MOF layers in solvents, and **3)** the reproducibility in multi-cycle gas sensing of patterned MOF diffractive sensors, as shown below.

1) Endurance in multilayered patterning. As shown in the examples of multilayered patterning, patterning of the 2nd or 3rd layer does not affect the integrity of the underlying layers (Fig. 4e, f). This can be ascribed to both the loss of solubility of cross-linked MOFs in the solvents used for the coating of the subsequent layers and the sufficient adhesion between MOF layers and the substrate.

In the revised work, we added SEM images of multilayered patterns covering an area of ~ 1 mm² (**Fig. R15b–d**). For instance, Fig. R15b shows uniform patterns of cross lines composed of ZIF-8@BrijC10 and HKUST-1@OA/OLAM in the whole area. For more quantitative analysis, we prepared two-layered binary MOF patterns, including those based on (first layer/second layer) ZIF-7@PEI/ZIF-8@BrijC10, ZIF-8@BrijC10/ZIF-7@PEI, and ZIF-8@BrijC10/HKUST-1@OA/OLAM. For ZIF-7@PEI/ZIF-8@BrijC10, the first layer contains ZIF-7@PEI, followed by the patterning of ZIF-8@BrijC10. In each case, we measured the thickness of the first layer before and after (in three different regions) the patterning of the second layer (**Fig. R16**). The heights of the first MOF layer measured at three different regions after patterning

the second layer are essentially the same with that before coating the second layer. These qualitative and quantitative results confirm the robustness/strong adhesion of the patterned MOF layers for subsequent patterning processes.

Figure R15. Optical images and EDS mapping of binary and ternary MOF patterns. (a) Cross-line patterns of ZIF-8@BrijC10 and HKUST-1@OA/OLAM. The linewidth is 50 μm . Inset in the upper part shows the magnified version highlighting the rectangles in dashed boxes (white, ZIF-8@BrijC10; red, HKUST-1@OA/OLAM). The bottom shows the corresponding EDS data. (b) Large-scale cross-line patterns of ZIF-8@BrijC10 and HKUST-1@OA/OLAM. The linewidth is 15 μm . (c) Large-scale patterns of alternating rectangles composed of ZIF-8@BrijC10 and HKUST-1@OA/OLAM. (d) Large-scale patterns containing three different MOFs (ZIF-8@BrijC10; HKUST-1@OA/OLAM; ZIF-7@PEI). Scale bars, 100 μm .

Figure R16. Optical images and height profiles of patterns of binary cross-line patterns of different MOFs. (a) ZIF-7@PEI/ZIF-8@BrijC10, (b) ZIF-8@BrijC10/ZIF-7@PEI, (c) ZIF-8@BrijC10/HKUST-1@OA/OLAM. In each case, the MOF written on the left was firstly patterned as lines, followed by the patterning of the second layer (those written on the right). The heights of the first MOF layer were measured before (the black traces) and after (the blue traces) the patterning of the second layer. The heights of the first layer after the second layer patterning were measured at three different locations. Scale bars, 1 mm and (insets) 500 μm .

2) Endurance of patterned MOF films during soaking tests. Another example showing the robustness of patterned MOF films on substrates is the soaking tests. In this case, we immersed the patterns of ZIF-8@BrijC10 in different solvents (acetone, ethanol, toluene, and methanol). The patterns remained intact after 24 h (**Fig. R17**).

Figure R17. Optical images of patterns of ZIF-8@BrijC10 after 24 h immersion in different solvents. (a) acetone, (b) ethanol, (c) toluene, (d) methanol. Scale bars, 500 μm .

μm , (in insets) $100\ \mu\text{m}$. According to ref.¹¹, the van der Waals attractive force between a single MOF NP and the flat substrate, at the separation of $1\ \text{nm}$, is about a few nN. The cross-linking between adjacent NPs in the patterns and the small separation between MOFs and substrates ($<1\ \text{nm}$) further increase the adhesion. This contributes to the good stability of patterned MOF films against solvent soaking.

3) Endurance of patterned MOF films during multi-cycle utilization. We used the diffraction grating vapor sensing as an example for their durability in multi-cycle utilization. The grating was immersed in dichloromethane solvent for 8 h prior to testing, and tested in 80% vapor pressure (P/P_0) of ethanol for 3 times. This completed one cycle of solvent immersion/gas vapor sensing. The grating was then immersed in dichloromethane for another 8 h before the vapor response test, for the second and third cycle. During this multi-cycle utilization, the optical power of first-order diffraction spot remained unchanged (**Fig. R18a**). After these tests, we measured the SEM images of the grating (Fig. R18b, c), where the MOF patterns were intact with well-defined facets. The small particles shown in Fig. R18b are probably from contaminants adsorbed on the grating sensor during measurements.

Figure R18. Responses and SEM images of the patterned ZIF-8@BrijC10 diffraction during multi-cycle utilization. (a) The optical power corresponding to the light intensity of the first-order diffraction spot in the 1st, 2nd, and 3rd cycle of measurements. The vapor pressure of ethanol for these measurements was 80%. (b, c) SEM images of the MOF patterns and the constituent MOF NPs in the diffraction sensor after 3 cycles of solvent immersion and gas vapor measurements. Scale bars, (b) $100\ \mu\text{m}$, (c) $500\ \text{nm}$, and (inset in c) $100\ \text{nm}$.

Our Changes:

1) We clarified the adhesion force between patterned MOF films and the substrates in the revised manuscript (Page 14–15).

“We further studied the adhesion stability and film retention curve of patterned MOF films. Note that the substrates used for patterning are not modified physically or chemically. Therefore, the adhesion between patterned MOF films and the substrates come mostly from the van der Waals forces. The van der Waals adhesion force can be estimated by equation in ref.⁵⁸ and be sufficient to hold MOF films firmly on substrates during the soaking test in different solvents for over 24 h (Supplementary Fig. 12).”

2) We added the data on the endurance of MOF films during soaking test (Fig. R17) as Supplementary Fig. 12 and related discussion in the revised Supplementary Information.

3) We added the data on the endurance of MOF films during consecutive, multilayered patterning (Figs. R15 and R16) as Supplementary Figs. 16 and 17 in the revised manuscript. We also added related discussions in the revised manuscript (Page 17).

“Due to the robust adhesion of MOF films on the substrates and the significantly reduced solubility of cross-linked MOF NPs, patterning of the second MOF layer does not affect the underlying cross-linked layer (Supplementary Fig. 16a–c).”

“To confirm the stability of the first patterned layer during the subsequent patterning procedures, we further compared the heights of patterned lines in the first layer before and after the patterning of the second layer, as shown in Supplementary Fig. 17. The heights of the first layer remain unchanged, regardless of the compositions of different layers.”

4) We added the data on the endurance of MOF grating gas sensors during multi-cycle utilization (Fig. R18) as Supplementary Fig. 24 in the revised manuscript. We also added related discussions in the revised manuscript (Page 23).

“We also tested the stability of such sensors in multicycle treatment and gas sensing (see details in Methods). During this multi-cycle utilization, the optical power of first-order diffraction spot remained unchanged (Supplementary Fig. 24a). After these tests, we measured the SEM images of the grating (Supplementary Fig. 24b, c), where the MOF patterns were intact with well-defined facets.”

5) We added the experimental details on the soaking test and the stability test of diffraction grating vapor sensors in “Methods” in the revised manuscript (Page 30–32).

“To check the endurance of patterned MOF films on substrates, we performed the soaking tests by soaking the patterned films in different solvents (acetone, ethanol, toluene, methanol) for 24 h and checked the optical images afterwards.”

“To check the durability of diffraction grating sensor in multi-cycle utilization, the grating was immersed in dichloromethane solvent for 8 h prior to testing, and tested in 80% vapor pressure (P/P_0) of ethanol for 3 times. This completed one cycle of solvent immersion/gas vapor sensing. The grating was then immersed in dichloromethane for another 8 h before the vapor response test, for the second and third cycle.”

Comment #3: Geometry. *Despite the novelty of the concept, I strongly recommend the Authors to indicate the existing (tens of nm) record values for MOF patterning (in the introduction), regardless of the method used and its complexity (Table S1). It is dangerous enough to mislead readers about achievements in this field, even if the*

information is correctly presented in the supplementary. The second point is the use of nanoMOFs as precursors for the pattern. Here I see 2 challenges: nanoMOFs possess a lower degree of porosity and a higher concentration of defects, which will negatively affect their sorption properties compared to an “ideal” large crystal; also, as the Authors found, the roughness of the patterns is related to the size of the nanoMOFs, and the smaller the MOF, the lower the roughness, while the more defects can be detected. I would like to see some discussion about this in the manuscript.

Our Response: We thank the reviewer for pointing this issue. We clarify the resolution of existing work on MOF patterning in the introduction of the revised manuscript and also added discussions on the concerns of nanoMOFs, as shown below.

1) The resolution of previous ebeam-based MOF patterning methods. We clarified in the introduction and the section of “Direct ebeam lithography of MOF films with nanoscale resolution” of the revised manuscript that existing methods (e.g., ref.¹⁶ by Ameloot group and ref.¹⁷ by Michael Tsapatsis group) based on ebeam lithography has enabled MOF patterning with resolution of sub-50 nm and ~100 nm, respectively, despite the requirement of high electron doses and other complexities described in Table S1.

2) Challenge of nanoMOFs on the low porosity and more defects. We agree with the reviewer that nanoMOFs are less ordered than their large crystal counterparts and contain more defects. However, these nanoMOFs can still show high porosity and BET surface area that are key characteristics of MOFs. According to previous reports^{5, 18, 19}, the BET surface area of colloidal nanoMOFs is in the range of ~900–1700 cm² g⁻¹, comparable to that of micro-sized MOFs. In our work, the BET surface area of various patterned MOF NP powers is also in this range. Detailed information appears in **Fig. R19** and **Table R2**.

Figure R19. N₂ adsorption–desorption isotherms and BJH size distribution of pristine and CLIP-MOF powders. (a–d) N₂ adsorption–desorption isotherms of (a) ZIF-8@BrijC10, (b) ZIF-8@CTAB, (c) UiO-66@PAA and (d) HKUST-1@OA. (e–h) Size distribution of (e) ZIF-8@BrijC10, (f) ZIF-8@CTAB, (g) UiO-66@PAA and (h) HKUST-1@OA.

Table R2. Summary of BET surface area, pore volume, and pore size of various MOF NP powders before and after CLIP-MOF treatment.*

MOFs	BET surface area (cm ² g ⁻¹)	total pore volume (cm ³ g ⁻¹)	micropore volume (cm ³ g ⁻¹)	pore diameter (nm)
ZIF-8@BrijC10	999.5	0.95	0.44	0.58
ZIF-8@BrijC10 CLIP-MOF	887.7	0.82	0.38	0.57
data in ref ⁴	1478.5	–	0.58	–
ZIF-8@CTAB	1721.3	1.22	0.63	0.87
ZIF-8@CTAB CLIP-MOF	1660.3	1.16	0.61	0.87
data in ref ⁵	1200	–	0.70	–
UiO-66@PAA	1209	1.13	0.49	0.57
UiO-66@PAA CLIP-MOF	1066	0.96	0.44	0.56
data in ref ⁶	1050	–	–	–
HKUST-1@OA	1400	1.11	0.57	0.43
HKUST-1@OA CLIP-MOF	1479	1.32	0.60	0.47
data in ref ⁷	1472	–	–	0.78

*Data in refs. show porosity data for MOF NPs synthesized in the same or similar approach and with similar sizes and ligands in previous reports.

Additionally, the structural defects and small sizes of nanoMOFs are sometimes advantageous, especially in applications relying on high surface area, incorporation of macromolecules, and fast mass transport^{3, 9, 20–24}. For example, the pores of MOFs are typically in the microporous range (<2 nm), which limits the access and transfer of macromolecules such as enzymes. By contrast, the arrangement of nanoMOFs can create mesopores that are suitable for the encapsulation of enzymes and the facile diffusion of substrates during enzymatic reactions²⁵.

More importantly, the use of colloidal MOF NPs permits the solution-based processing of MOF films and the direct patterning based on light- or ebeam irradiation triggered

solubility changes. The obtained patterned MOFs maintain good porosity and BET surface area, which allows their use in prototype vapor sensors. The unique features of colloidal nanoMOFs such as the facile mass transport and the good compatibility with biomolecules and drugs²⁶ may also lead to more possibilities for patterned MOFs in the fields of photonics, biosensing, biomedical applications, and catalysis.

3) Challenge of nanoMOFs on surface roughness of the patterned MOF films. The roughness of the patterned films is related to the size of the MOF NPs; the smaller the MOF nanocrystal, the lower the roughness. However, as described above, the small MOF NPs may still show decent porosity. For instance, the roughness of patterned ZIF-8@BrijC10 NP film is about 13 nm (Fig. 3f), which is smaller than the NP size (~30 nm) and on par with that of halogen ZIFs patterned by direct ebeam lithography in previous report¹⁶. At the same time, these patterned ZIF-8@BrijC10 NPs show BET surface area of ~900 cm² g⁻¹. Other patterned MOFs (e.g., ZIF-8@CTAB) can even show BET surface area of over 1600 cm² g⁻¹. We included these potential limitations and discussions in the Discussion section of the revised manuscript.

Our Changes:

1) The resolution of previous ebeam-based MOF patterning methods. We clarified in the introduction and the section of “Direct ebeam lithography of MOF films with nanoscale resolution” that existing methods based on ebeam lithography can achieve nanoscale resolution (e.g., sub-50 nm).

Page 4, “The switched solubility after irradiation allows for the selective dissolution of ZIFs in the exposed regions, producing well-defined, micro- and nanoscale patterns. The patterning resolution under ebeam irradiation has reached sub-50 nm while the porous structures are maintained.”

Page 24, “These chemical/structural modifications lead to solubility changes of irradiated MOFs in certain solvents and support MOF patterning. The patterning resolution can achieve sub-50 nm.”

2) We added the data on the BET analysis of various MOF NPs before and after patterning (or treated by the procedures in CLIP-MOF) in Fig. 5, Supplementary Fig. 21, and Supplementary Table 3 in the revised manuscript and Supplementary Information.

3) We added discussions on the potential limitation of using nanoscale MOF NPs in the patterning in the Discussion section of the revised manuscript (Page 27–28).

“One should note that the use of colloidal MOF NPs is also associated with challenges and opportunities related to the small crystal size, defects, and the packing fashion. Recent advancements in colloidal MOF NPs^{43, 60, 69–72} allow CLIP-MOF to benefit from

nanoscale MOFs with not only decent porosity and surface area but also unique features for biosensing and catalysis, such as defect-related catalytic activity and packing-induced mesoporous structures to host macromolecules. The sizes of MOF NPs also affect the surface roughness of patterned MOFs in CLIP-MOF; smaller NPs with uncompromised porosity are preferred for patterning smooth films.”

Reviewer #3 (Remarks to the Author):

Comment: *I have read and considered the paper by Tian et al. submitted for publication. The authors report on the use of a ligand cross-linking chemistry combined with a variety of different metal organic frameworks (MOFs) that behaves as a negative-tone photo- and electron-beam lithography resist. Importantly they show that, at least for some cases, the MOFs maintain their functional properties through a sensing/electrochromic experiment. The paper is well structured, the many experimental results are solid and the discussion always appropriate.*

Our Response: We thank the review for the positive comments. In the revised manuscript, we specified the cross-linker amount in the applications shown in Fig. 5, added the contrast curve of the patterning of ZIF-8 MOFs, and clarified the aging time of MOF NP inks.

Comment #1: *Can you specify the amount of cross-linker used in the solutions of Fig. 5? This is particularly relevant as these patterns were tested to verify the MOFs functionality.*

Our Response: We thank the reviewer for the comments. In the revised manuscript, we modified Fig. 5 to include the XRD patterns, BET analysis, and other characterization data for various MOFs before and after patterning. We also made a new figure (Fig. 6) to show the data on diffraction grating vapor sensor and electrochromic devices. We specified the amount of cross-linkers used for these experiments (i.e., those involved in Figs. 5 and 6).

Patterned films for XRD measurements were made from a solution containing 5–20 wt% of the cross-linkers (to the mass of MOFs, as summarized in Supplementary Table 2). The diffraction grating sensors and pixelated electrochromic devices were made from a solution containing ZIF-8@BrijC10 and bisPFPA cross-linkers (the mass ratio of bisPFPA to ZIF-8@BrijC10 \approx 10 wt%). For BET analysis, we irradiated a MOF solution containing bisPFPA cross-linkers (the mass ratio of bisPFPA to ZIF-8@BrijC10 \approx 20 wt%) with UV light for an extended period (\sim 5 min, corresponding to 900 mJ cm⁻²) to ensure complete cross-linking. The MOF NP powders were collected for BET analysis.

Our Changes: We added this information in the revised manuscript.

Page 19, “As shown in the XRD data in Fig. 5a–e and Supplementary Fig. 18, MOF NPs including ZIF-8@BrijC10, ZIF-8@CTAB, ZIF-7@PEI, UiO-66@PAA, HKUST-1@OA, HKUST-1@OA/OLAM, and Eu(BTC)@PVP maintain their crystal structure and crystallinity after being treated by the CLIP-MOF patterning procedures (addition of cross-linkers with the amount of 5–20 wt% to the mass of MOFs, UV exposure, and removal of unreacted cross-linkers by developer solvents).”

Page 34, “For MOF NPs treated with CLIP-MOF procedures, the MOF solution containing bisPFPA cross-linkers (up to 20 wt% to the mass of MOF NPs) were exposed to UV for extended period (~5 min, corresponding to 900 mJ cm⁻²) to ensure complete cross-linking.”

Page 31, “**Procedures for Gas Sensing.** The diffraction grating structures were patterned with ZIF-8@BrijC10 MOF NPs (with ~10 wt% of bisazide cross-linkers in the coating solution).”

Page 32, “The precipitates of MOF NPs were collected and redispersed in chloroform. This MOF NP solution was mixed with bisPFPA cross-linkers (~10 wt% to the mass of MOF NPs) for patterning via CLIP-MOF.”

Comment #2: *Did you perform any contrast curve at least of one of the resist formulations? I think it would add value to the paper.*

Our Response: We thank the review for the very helpful comment. The contrast curve or film retention was measured via inductively coupled plasma-optical emission spectroscopy (ICP-OES) analysis of Zn. We exposed ZIF-8@BrijC10 NP films (exposure in the entire film) under different UV doses. After solvent developing, we collected the MOF NPs dissolved in the developer by evaporation. We then digested 1) the MOF NPs remaining in the film and 2) MOF NPs collected from the developer by using concentrated hydrochloric acid. By comparing the Zn concentration in these two digested solutions, we can calculate the ratio of MOF NPs remaining in the film after UV exposure and plot the film retention/contrast curve (**Fig. R20**). A low dose of 60 mJ cm⁻² results in over 60% film retention that subsequently saturates (~83%) at 90 mJ cm⁻². The non-zero film retention for unexposed films may come from nonspecific adhesion of residual MOF NPs after solvent developing.

Figure R20. Film retention or contrast curve of ZIF-8@BrijC10 films at different UV doses. A low dose of 60 mJ cm⁻² results in over 60% film retention that subsequently saturates (~83%) at 90 mJ cm⁻². The non-zero film retention for unexposed films may come from nonspecific adhesion of residual MOF NPs after solvent developing.

Our Changes:

1) We added Fig. R20 as Supplementary Fig. 13 in the revised Supplementary Information.

2) We added related descriptions in the revised manuscript (Page 15).

“For more quantitative estimation of the patterning efficacy, we measured the film retention or contrast curve of ZIF-8@BrijC10 MOF films at different UV doses (Supplementary Fig. 13). The film retention reaches ~83% at the UV dose of ~90 mJ cm⁻².”

3) We added the method for estimating the film retention and contrast curve data in the revised Methods (Page 33).

“The contrast curve or film retention was measured via inductively coupled plasma-optical emission spectroscopy (ICP-OES) analysis of Zn. We exposed ZIF-8@BrijC10 NP films (exposure in the entire film) under different UV doses. After solvent developing, we collected the MOF NPs dissolved in the developer by evaporation. We then digested 1) the MOF NPs remaining in the film and 2) MOF NPs collected from the developer by using concentrated hydrochloric acid. By comparing the Zn concentration in these two digested solutions, we can calculate the ratio of MOF NPs remaining in the film after UV exposure and plot the film retention/contrast curve.”

Comment #3: *Can you comment on the aging time of the resist? I mean, for how many days/weeks the solutions are stable and contrast curves (or processing parameters) are comparable?*

Our Response: We thank the reviewer for the comments. Because the patterning relies on the solubility contrast of MOF NPs before and after UV exposure, freshly made MOF NP solutions with great colloidal stability are preferred. One can also use MOF solutions after stored for several days as long as they remain colloidally stable.

Following the reviewer's suggestion, we evaluated the aging time of the mixed solutions of bisPFPA and ZIF-8@BrijC10 NPs by checking the film retention. In brief, freshly made solution of ZIF-8@BrijC10 (50 mg mL⁻¹) was added with bisPFPA (~10 wt% to the mass of ZIF-8@BrijC10) and stored under dark condition. We used this solution for patterning and measured the film retention by using ICP-OES analysis (as shown in the response to Comment #2 of this reviewer) on day 0, day 10, day 15, and day 30, respectively (**Fig. R21**). When kept for 10 days, the film retention dropped very slightly from 80% to 78%, which is negligible considering the measurement errors. In comparison, patterning with a solution stored for 15 or 30 days showed a notably decreased film retention of 62%, even though there was no obvious destabilization of the solution (inset in Fig. R21). The decreased patterning performance may come from

the partial aggregation and the formation of small clusters of MOF NPs. These results suggest the aging time for ZIF-8@BrijC10 is about 10 days for best performance in patterning. In our work, colloidal MOFs and cross-linkers are stored separately and mixed at the time of use. Cross-linkers can be stored in dark for more than six months and colloidal MOFs for more than one month, which is determined by the colloidal stability of MOF NP solutions and dependent on the NP sizes, ligands, concentration, and others.

Figure R21. Film retention of ZIF-8@BrijC10 MOF films versus aging days. Inset shows the photos of solutions containing ZIF-8@BrijC10 and bisazides after stored in dark for different days. We monitored the film retention by using ICP-OES analysis with MOF NP solutions after stored for 0, 10, 15, and 30 days. When stored for 10 days, the film retention dropped very slightly from 80% to 78%, which is negligible considering the measurement errors. In comparison, patterning with a solution stored for 15 or 30 days showed a notably decreased film retention of 62%, even though there was no obvious destabilization of the solution (inset in Fig. R21). The decreased patterning performance may come from the partial aggregation and the formation of small clusters of MOF NPs. These results suggest the aging time for ZIF-8@BrijC10 is about 10 days for best performance in patterning. In our work, colloidal MOFs and cross-linkers are stored separately and mixed at the time of use. Cross-linkers can be stored in dark for more than six months and colloidal MOFs for more than one month, which is determined by the colloidal stability of MOF NP solutions and dependent on the NP sizes, ligands, concentration, and others.

Our Changes:

1) We added Fig. R21 as Supplementary Fig. 14 in the revised Supplementary Information.

2) We added related descriptions in the revised manuscript (Page 15).

“From the film retention data, we also estimated the storage/aging time for the solution containing MOF NPs and cross-linkers (Supplementary Fig. 14). Because the patterning relies on the contrast in colloidal solubility of MOFs before and after UV exposure, freshly made MOF solutions with great colloidal stability are preferred and the aging time is about 10 days.”

Reference

1. Papurello, R.L. et al. Post-synthetic modification of ZIF-8 crystals and films through UV light photoirradiation: Impact on the physicochemical behavior of the MOF. *ChemPhysChem* **20**, 3201–3209 (2019).
2. Park, K.S. et al. Exceptional chemical and thermal stability of zeolitic imidazolate frameworks. *PNAS* **103**, 10186–10191 (2006).
3. Wu, H. et al. Unusual and highly tunable missing-linker defects in zirconium metal–organic framework UiO-66 and their important effects on gas adsorption. *J. Am. Chem. Soc.* **135**, 10525–10532 (2013).
4. He, M. et al. Synthesis of zeolitic imidazolate framework-7 in a water/ethanol mixture and its ethanol-induced reversible phase transition. *ChemPlusChem*, **78**, 1222–1225 (2013).
5. Zhao, X., Fang, X., Wu, B., Zheng, L. & Zheng, N. Facile synthesis of size-tunable ZIF-8 nanocrystals using reverse micelles as nanoreactors. *Sci. China Chem.* **57**, 141–146 (2014).
6. Pan, Y., et al. Tuning the crystal morphology and size of zeolitic imidazolate framework-8 in aqueous solution by surfactants. *CrystEngComm* **13**, 6937–6940 (2011).
7. Morris, W. et al. Role of modulators in controlling the colloidal stability and polydispersity of the UiO-66 metal-organic framework. *ACS Appl. Mater. Inter* **9**, 33413–33418 (2017).
8. Cai, X., Xie, Z., Pang, M. & Lin, J. Controllable synthesis of highly uniform nanosized HKUST-1 crystals by liquid–solid–solution method. *Cryst. Growth*

- Des.* **19**, 556–561 (2019).
9. Jeong, U. Inversion of dispersion: Colloidal stability of calixarene-modified metal–organic framework nanoparticles in nonpolar media. *J. Am. Chem. Soc.* **141**, 12182–12186 (2019).
 10. Espín, J., Garzón-Tovar, L., Carné-Sánchez, A., Imaz, I. & MasPOCH D. Photothermal activation of metal–organic frameworks using a uv–vis light source. *ACS Appl. Mater. Inter* **10**, 9555–9562 (2018).
 11. J. N. Israelachvili, *Intermolecular and Surface Forces*. (ed. Third, 2011).
 12. Yang, L., Wen, J. Can DLVO theory be applied to MOF in different dielectric solvents? *Micropor. Mesopor. Mater* **343**, 112166 (2022).
 13. Wang, Y., Fedin, I., Zhang, H. & Talapin D.V. Direct optical lithography of functional inorganic nanomaterials. *Science* **357**, 385–388 (2017).
 14. Lu, S. et al. Beyond a linker: The role of photochemistry of crosslinkers in the direct optical patterning of colloidal nanocrystals. *Angew. Chem. Int. Ed.* **61**, e202202633 (2022).
 15. Fu, Z. et al. Direct photo-patterning of efficient and stable quantum dot light-emitting diodes via light-triggered, carbocation-enabled ligand stripping. *Nano Lett.* **23**, 2000–2008 (2023).
 16. Tu, M. et al. Direct X-ray and electron-beam lithography of halogenated zeolitic imidazolate frameworks. *Nat. Mater.* **20**, 93–99 (2021).
 17. Miao, Y. et al. Solvent-free bottom-up patterning of zeolitic imidazolate frameworks. *Nat. Commun.* **13**, 420 (2022).
 18. Cravillon, J. et al. Controlling zeolitic imidazolate framework nano- and

- microcrystal formation: Insight into crystal growth by time-resolved in situ static light scattering. *Chem. Mater* **23**, 2130–2141 (2011).
19. Demessence, A. et al. Adsorption properties in high optical quality nanoZIF-8 thin films with tunable thickness. *J. Mater. Chem.* **20**, 7676–7681 (2010).
 20. Schaate, A. et al. Modulated synthesis of zr-based metal–organic frameworks: from nano to single crystals. *Chem. Eur. J* **17**, 6643–6651 (2011).
 21. Decker, G.E., Stillman, Z., Attia, L., Fromen, C.A. & Bloch, E.D. Controlling size, defectiveness, and fluorescence in nanoparticle UiO-66 through water and ligand modulation. *Chem. Mater.* **31**, 4831–4839 (2019).
 22. Daniel, M., Mathew, G., Anpo, M. & Neppolian, B. MOF based electrochemical sensors for the detection of physiologically relevant biomolecules: An overview. *Coord. Chem. Rev.* **468**, 214627 (2022).
 23. Peng, S. et al. Metal-organic frameworks for precise inclusion of single-stranded DNA and transfection in immune cells. *Nat. Commun* **9**, 1293 (2018).
 24. Morris, R.E. & Brammer, L. Coordination change, lability and hemilability in metal–organic frameworks. *Chem. Soc. Rev.* **46**, 5444–5462 (2017).
 25. Zhao, B., et al. Ferroptosis-mediated synergistic therapy of hypertrophic scarring based on metal–organic framework microneedle patch. *Adv. Funct. Mater.* **33**, 2300575 (2023).
 26. Troyano, J., Carné-Sánchez, A., Avci, C., Imaz, I. & Maspoch, D. Colloidal metal–organic framework particles: the pioneering case of ZIF-8. *Chem. Soc. Revi.* **48**, 5534–5546 (2019).

REVIEWERS' COMMENTS

Reviewer #1 (Remarks to the Author):

First of all, the authors did a very good job by adding all results and I was surprised to see how structural stable are the MOF. The only not stable combination is HKUAT-1@OA/OLAM and hence mention this in the text and maybe why this material is unstable but HKUST-1@OA not. The strength of this study is that they show different MOFs as processability is a key issue in the MOF field. All my comments are address and reading the other reviewers I think they did the same. As reviewer 2 mention the concept is not new but the material class is and hence the impact for Nature Communication is for me there. Therefore, I recommend acceptance of this article and I would to ask the authors always carefully characterize the material in order to back up claims.

Reviewer #2 (Remarks to the Author):

Revised version of the manuscript corresponds to the journal level.

Reviewer #3 (Remarks to the Author):

The authors have correctly addressed my concerns/questions in in the revised manuscript.

Response to Reviews

Reviewer #1 (Remarks to the Author):

Comment: *First of all, the authors did a very good job by adding all results and I was surprised to see how structural stable are the MOF. The only not stable combination is HKUST-1@OA/OLAM and hence mention this in the text and maybe why this material is unstable but HKUST-1@OA not. The strength of this study is that they show different MOFs as processability is a key issue in the MOF field. All my comments are addressed and reading the other reviewers I think they did the same. As reviewer 2 mention the concept is not new but the material class is and hence the impact for Nature Communications is for me there. Therefore, I recommend acceptance of this article and I would to ask the authors always carefully characterize the material in order to back up claims.*

Our Response: We really appreciate the reviewer's comments. These comments, especially those on complete characterizations of patterned MOFs, have greatly helped us to improve the quality of our work.

Figure R1. SEM images of HKUST-1@OA/OLAM films patterned by CLIP-MOF. Scale bars, 1 μm and 100 nm (for insets).

Regarding the issue of HKUST-1@OA/OLAM, we would like to clarify that these MOFs are also stable during the patterning. This can be supported by the unchanged morphology of patterned HKUST-1@OA/OLAM, as shown in Supplementary Figure 15 in the Supplementary Information (or **Fig. R1** here). The SEM images and height profiles of binary or ternary patterns containing HKUST-1@OA/OLAM layers (Supplementary Figures 16 and 17) also suggest the preserved structures of these MOFs.

However, due to the small size (about 3 nm), both pristine and patterned HKUST-1@OA/OLAM show lower crystallinity than their counterparts with large crystal sizes (e.g., HKUST-1@OA with average size of about 80 nm) and also broad peaks in powder X-ray diffraction patterns. We included this description in the caption of Supplementary Figure 18.

Our Changes: As suggested by the reviewer, we added explanations on the relatively

broad powder XRD patterns in the revised manuscript (Page 19) to avoid confusion.

“No impurity peaks, additional peak widening, or changes in the relative peak intensities were observed after patterning. Note that the relatively broad peaks of pristine and patterned HKUST-1@OA/OLAM are due to their small particle sizes (about 3 nm, as estimated by Scherrer equation, Supplementary Fig. 18). HKUST-1 synthesized with only OA ligands have larger particle sizes and hence more evident diffraction peaks. Nonetheless, both HKUST-1@OA and HKUST-1@OA/OLAM preserve their structures and crystallinity after patterning.”

Reviewer #2 (Remarks to the Author):

Comment: *Revised version of the manuscript corresponds to the journal level.*

Our Response: We thank the reviewer for this and previous comments, which have helped us to improve our manuscript!

Reviewer #3 (Remarks to the Author):

Comment: *The authors have correctly addressed my concerns/questions in in the revised manuscript.*

Our Response: We thank the reviewer for this and previous comments, which have helped us to improve our manuscript!